# HYPERTILING – a high performance Python library for the generation and visualization of hyperbolic lattices

Manuel Schrauth[1,2*], Yanick Thurn[1], Florian Goth[1], Jefferson S. E. Portela[1],
Dietmar Herdt[1] and Felix Dusel[1,3]

**1** Julius-Maximilians-Universität Würzburg (JMU), Institute for Theoretical Physics and
Astrophysics, Würzburg, Germany
**2** Fraunhofer Institute for Integrated Circuits (IIS), Erlangen, Germany
**3** University of British Columbia (UBC), Vancouver, Canada

\* manuel.schrauth@physik.uni-wuerzburg.de

June 27, 2024

## Abstract

HYPERTILING is a high-performance Python library for the generation and visualization
of regular hyperbolic lattices embedded in the Poincaré disk model. Using highly op-
timized, efficient algorithms, hyperbolic tilings with millions of vertices can be created
in a matter of minutes on a single workstation computer. Facilities including computa-
tion of adjacent vertices, dynamic lattice manipulation, refinements, as well as powerful
plotting and animation capabilities are provided to support advanced uses of hyperbolic
graphs. In this manuscript, we present a comprehensive exploration of the package, en-
compassing its mathematical foundations, usage examples, applications, and a detailed
description of its implementation.

# 1 Introduction

The exploration of curved spaces is motivated by the recognition that the intrinsic geometry of a system can dramatically affect its behavior and characteristics, and that the flat Euclidean geometry, which accurately describes our everyday experiences, is not universally applicable across all scales and contexts. Curvature plays a significant role in various branches of science, most notably general relativity. Although measurements of the cosmic microwave indicate that the universe as a whole is flat or very close to it [1–3], curvature remains a key concept

in cosmology and astronomy, as massive objects directly change space and time around them. Spaces with negative curvature in particular are of great interest. These so-called *hyperbolic spaces* play a decisive role in the AdS/CFT correspondence [4–7], which provides a duality between conformal field theory (CFT) operators and Anti-de Sitter (AdS) gravity fields and is of great significance both for fundamental aspects of quantum gravity [8] and for applications to strongly correlated condensed matter systems [9].

Applications of curved manifolds are also found in many other fields of science and engineering, for instance, in large-scale climate simulations encompassing the entire planet Earth. Accounting for the effects of curvature becomes then crucial for accurately modeling and predicting weather patterns, climate changes and their potential impact on ecosystems. Specifically, Earth can be represented as a two-sphere, $\mathbb{S}_2$, which is a compact manifold of constant positive curvature. Constant *negative* curvatures, on the other hand, correspond to hyperbolic spaces. These non-compact manifolds distinguish themselves from flat spaces, in that, for instance, the volume encompassed by a ball of radius $r$ grows exponentially with $r$ instead of polynomially, and there are not one, but infinitely many parallels to any given line $L$, passing through any point $P$ not on $L$.

Manifolds of constant curvature are fully characterized by their scalar curvature radius $\ell$. From a more abstract point of view, $\ell$ can be seen as a control parameter, representing a natural extension of the usual flat geometry. Induced by the curvature, established physical systems exhibit different behaviors or even entirely new phenomena, which, in turn, can be used as probes, yielding novel insights into the physics of the corresponding flat models, in the limit $\ell \to \infty$. Examples of physical processes which are substantially affected by their supporting geometry can be found in diffusive systems [10–14], in magnetic properties of nano-devices [15–17], soft materials [18], complex networks [19, 20], including information infrastructure [21], quantum gravity [22–24], bio-membranes [11], glass transitions [25, 26], equilibrium spin systems and critical phenomena [27–30] as well as adsorption and coating phenomena on non-flat surfaces [26].

The investigation of many of the phenomena mentioned so far demands a numerical approach, often involving a discretized geometric representation, which is generally a nontrivial task in hyperbolic spaces. The purpose of HYPERTILING [31] is to provide a robust, powerful and flexible solution for this step.

This paper is organized as follows: In the remainder of this Introduction, we briefly present hyperbolic spaces and lattices, their application and existing numerical implementations. In Section 2, we show how easy it is to install and use HYPERTILING and in Section 3 we showcase the range of features it offers. Section 4 is a short summary of the package's underlying mathematical foundations and Section 5 provides an extensive discussion of our algorithmic numerical implementations, their different features and performances. We show the package in action in Section 6, discuss our future plans in Section 7 and offer our closing remarks in Section 8.

## 1.1 Hyperbolic Lattices

A manifold equipped with constant negative curvature (a hyperbolic space) is commonly denoted by the symbol $\mathbb{H}^d$, where $d$ is the number of spatial dimensions. Formally, it can be embedded in a $d + 1$ dimensional space with Minkowskian signature. Specifically, it constitutes the hypersurface constrained by the relation

$$x^\mu x_\mu = \eta_{\mu\nu} x^\mu x^\nu = -x_0^2 + x_1^2 + \ldots + x_d^2 = -\ell^2, \tag{1}$$

where $\eta = \mathrm{diag}(-1, 1, \ldots, 1)$. The line element is given by

$$\mathrm{d}s^2 = -\mathrm{d}x_0^2 + \mathrm{d}x_1^2 + \ldots\ldots + \mathrm{d}x_d^2. \tag{2}$$

In this paper, we restrict ourselves to the case $d = 2$, also known as the *pseudosphere*. This notion stems from an apparent resemblance of the $\mathbb{H}_2$ metric to that of an ordinary sphere. Using the parametrization $x_0 = \cosh\rho$, $x_1 = \sinh\rho\cos\phi$ and $x_2 = \sinh\rho\sin\phi$, we arrive at

$$\mathrm{d}s^2 = \mathrm{d}\rho^2 + \sinh^2\rho\,\mathrm{d}\phi^2, \tag{3}$$

where $\rho \in [0, \infty)$ and $\phi \in [0, 2\pi)$. In the literature, a certain variety of similar, polar-like coordinates representations can be found, a frequently used one being

$$\mathrm{d}s^2 = \frac{1}{1 + r^2/\ell^2}\mathrm{d}r^2 + r^2\mathrm{d}\phi^2, \tag{4}$$

where the curvature radius $\ell$ enters explicitly and $r \in [0, \infty)$. All these coordinate systems are mathematically equivalent in that they describe the same manifold. The coordinate representation primarily used in this paper is the so-called *Poincaré disk model* of the hyperbolic space, denoted as $\mathbb{D}_2$. Its metric is given by

$$\mathrm{d}s^2 = \frac{4\ell^2}{(1 - z\bar{z})^2}\mathrm{d}z\mathrm{d}\bar{z}, \tag{5}$$

where $z \in \mathbb{C}$, $|z| < 1$. Note that $K = -1/\ell^2$ represents the constant negative curvature of the manifold. In the Poincaré model, the entire $\mathbb{H}_2$ space is projected onto the complex plane, with the unit circle representing points infinitely far away from the origin.

The hyperbolic two-space can be naturally discretized by *regular hyperbolic tilings* [32], which have been studied already since the late 19th century [33, 34]. They become known to a broader scientific audience due to the works of H.S.M. Coxeter [35, 36], which also have been an inspiration for M.C. Eschers famous Circle Limit drawings [37]. Regular hyperbolic tilings preserve a large subgroup of the isometries of $\mathbb{H}_2$ [38, 39], which makes them promising candidates for a wide range of numerical simulations setups. Regular tilings are characterized by their Schläfli symbol $(p, q)$, where the condition $(p-2)(q-2) > 4$ has to be met in order for a tiling with $q$ regular $p$-gons meeting at each vertex, to be hyperbolic. The $(7, 3)$ hyperbolic tiling and its dual $(3, 7)$ tiling are shown in Figure 1 as an example. Since hyperbolic spaces exhibit a length scale, defined by their radius of curvature $\ell$, the edge lengths of hyperbolic polygons are fixed quantities, depending only on the Schläfli parameters $p$ and $q$ [40]. Their geodesic length $h^{(p,q)}$ in units of $\ell$ can be computed via the Poincaré metric (5) and can be interpreted as a fixed lattice spacing that cannot be tuned[1]. In general, the inherent length scale has significant implications for the discretization of hyperbolic spaces. Foremost, it renders a continuum limit of $(p, q)$ tilings in the usual way impossible, which severely limits the applicability of traditional finite-size scaling methods [41, 42].

Scaling, i. e. shrinking or enlarging a regular polygon in a hyperbolic or spherical space will also influence its shape, evidenced by changes in the interior angles at the vertices, and the regular tessellation will in general no longer cover the space without voids or overlaps. Technically speaking, there is no concept of *similarity* in curved spaces. Moreover, these geometric peculiarities render the construction of periodic boundaries – which can be indispensable when studying bulk systems due to the generically *large* boundary of hyperbolic spaces – particularly challenging [43]. Broadly speaking, the bounding edges of a suitable finite part of the tessellation need to be glued together properly in order to obtain a translationally invariant lattice. A systematic approach of how compact surfaces can be tessellated is given by the theory of so-called *regular maps* [44–47]. Just like other compact surfaces, regular maps can be embedded in 3D Euclidean space, such as beautifully demonstrated in Reference [48].

---

[1]A detailed discussion can be found Section 4.3.

## 1.2  Applications

Hyperbolic lattices found particular interest in the field of critical phenomena [49, 50] over the last two decades. For the Ising model [51] there are strong indications that the critical exponents take on their corresponding mean-field values as the hyperbolic grid can be regarded as effectively infinite-dimensional [52, 53]. Moreover, even at very high temperatures small-sized ferromagnetic domains can be observed [54]. Despite the mean-field properties on hyperlattices, the correlation length does not diverge at criticality, rather it stays finite, thus indicating the existence of an inherent length scale linked to the curvature radius which destroys the usual concept of scale invariance at criticality [55, 56]. Also, other equilibrium critical phenomena have been examined on hyperbolic lattices, including the $q$-state Potts model and the $XY$ model [25, 57, 58]. For the latter, it turned out that the hyperbolic surface induces a zero-temperature glass transition even in systems without disorder. This is due to the non-commutativity of parallel transport of spin vectors which causes a breakdown of their perfect orientational order and consequently gives rise to local frustration. Even more striking novel effects were found in *percolation* systems on hyperbolic lattices [59–65]. Specifically, an intermediate phase associated with two critical thresholds arises. At the lower critical probability, infinitely many unbounded clusters emerge. At the upper critical point, these clusters join into one unique unbounded cluster, spanning the entire system. In the flat Euclidean limit, these two thresholds coincide and the intermediate phase vanishes. It was found that this behavior is due to the non-vanishing surface-volume ratio of these lattices in the infinite-volume limit.

Besides critical phenomena, the physics of hyperbolic tilings has recently been studied in the context of condensed matter physics [66], circuit quantum electrodynamics [67–69], quantum field theory [70, 71] and topolectric circuits [72–74]. Another research branch where hyperbolic lattices arise very naturally is the AdS/CFT correspondence [4], as an Anti de-Sitter space with Euclidean signature, EAdS$_2$, is isomorphic to $\mathbb{H}_2$. Current attempts to discretize the AdS/CFT correspondence are based on modeling hyperbolic spaces [75]. Very recently, some of the authors were able to show that the Breitenlohner-Freedman bound [76, 77], a central result in supergravity, which states that certain perturbations that are unstable in flat geometries are actually stable on hyperbolic spaces, thus allowing for a straightforward experimental realization via hyperbolic electric circuits [78]. In this study, an earlier version of the HYPERTILING package was used.

## 1.3  Existing Implementations

The paper *Hyperbolic Symmetry* [79] by Douglas Dunham has been a very influential work in the numerical exploration of regular hyperbolic geometry. Implementations soon after its publication [80] up until today [81] are based on this algorithm. Scientific applications such as those described in Section 1.2, especially with a numerical focus, demand frameworks for constructing hyperbolic lattices and, over the years, individual researchers and small research groups have been developing their own codes. These codes, however, are typically neither openly available nor maintained after the publication of their associated work and are therefore of little use for the wider research community. Among the implementations that are available, some are for demonstration or educational purposes and have as their main objective simply displaying hyperbolic tilings [82–86], sometimes with artistic goals [81, 87], or supporting manufacturing applications [88, 89] that demand the construction of relatively small tessellations. Other, more group-theoretic approaches, are understood as proof of concept rather than high performance modules [90, 91]. Hence, all these projects do not provide the performance, data availability, facilities/resources and documentation required for sustained scientific research. It is with the aim of fulfilling these research needs that HYPERTILING has been created.

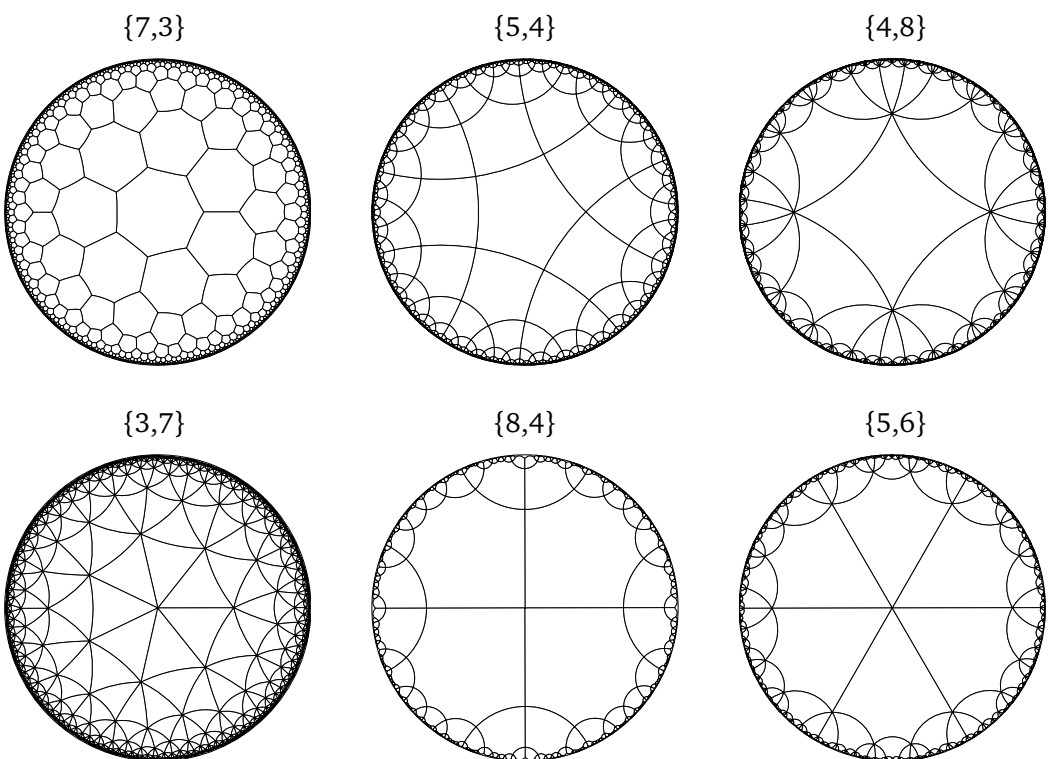

Figure 1: Selection of regular hyperbolic tilings projected onto the Poincaré disk. Tilings in the upper (lower) row are centered about a cell (vertex).

## 2  Setup

### 2.1  Environment

HYPERTILING is a Python package and should run everywhere where a Python 3 interpreter is available. In order to construct and visualize basic hyperbolic lattices, we only require two very common package dependencies, namely *numpy* and *matplotlib*. To fully utilize the high performance aspect of the library, we furthermore recommend installing *numba*, which is used to significantly speed up many of HYPERTILING's internal functions. However, note that even without *numba* the package is fully functional, only potentially slower. Finally, specific optional dependencies are the package *sortedcontainers*[2], employed for a faster internal memory layout of specific construction kernels, as well as *networkx*[3], which can be a useful extension of the visualization capabilities already provided directly in HYPERTILING. Note that all these packages are available via standard sources, such as PyPI or conda.

### 2.2  Installation

The HYPERTILING library can be installed directly from the PyPI package index using the ubiquitous pip installer:

```
python -m pip install hypertiling
```

All releases as well as the latest version can also be downloaded or cloned from our public GitLab repository [92], using

---

[2]https://grantjenks.com/docs/sortedcontainers
[3]https://networkx.org

```
git clone https://git.physik.uni-wuerzburg.de/hypertiling/
    hypertiling.git
```

For a local installation then from its root directory execute

```
python -m pip install .
```

## 2.3  Quick Start

After successful installation, the package can be imported into any Python 3 interpreter and a first plot of a hyperbolic lattice is readily created with only few lines of code:

```python
from hypertiling import HyperbolicTiling
from hypertiling.graphics.plot import quick_plot

p, q, n = 7, 3, 4

tiling = HyperbolicTiling(p, q, n)

quick_plot(tiling)
```

This should display a tessellation similar to the (7,3) lattice shown in Figure 1. In the code example, the parameter *n* denotes the number of layers of the tilings, a concept which relates to the size of the tiling and is also termed *coronas* or *generations* in the literature.

A couple of further examples of the capabilities of the package are found in the remainder of this manuscript, however, a larger selection of code examples and use cases is available as interactive Jupyter notebooks in the code base.

In the following sections, we explore the features of the package in more detail. In order to keep the document concise, the complete in-depth documentation and API reference of the package is included in the code repository https://git.physik.uni-wuerzburg.de/hypertiling/hypertiling [92].

## 3  Features

The HYPERTILING package centers on the construction of tilings of the two-dimensional hyperbolic plane. A hyperbolic tiling (or *tessellation*) consists of individual *cells* or, in two dimensions, *polygons*, which are arranged to cover the hyperbolic manifold without voids or overlappings. We use the terms cells/polygons and tiling/tessellation/lattice interchangeably. Since particular focus is placed on *regular* tilings, it is common to identify tilings by their Schläfli symbol $(p, q)$, with $(p - 2)(q - 2) > 4$ for hyperbolic curvature. In a regular tiling, all $p$-gonal cells are identical/uniform in the sense of geometric congruence. For a selection of visualizations, we refer the reader to Figure 1. Besides hyperbolic tilings, HYPERTILING also offers the functionality to construct *graphs*, which can be interpreted as reduced, coordinate-free lattices. They comprise only the adjacency relations between vertices.

Constructing tilings and associated graphs represents the core functionality of HYPERTILING, which the library makes particularly simple, as shown in Section 2.3. For more advanced use, we first need to introduce the concept of *kernels*. In HYPERTILING, a kernel encodes the algorithmic construction, the data structure and certain peripheral methods and auxiliary functions of a tiling or a graph. However, since we stick to a uniform user interface, irrespective of the kernel used, greatest possible flexibility is ensured and the library can be used with only little to no knowledge about technical, kernel-specific details. Our interfaces are held transparent and the user can benefit from the capabilities of the different kernels without any detailed knowledge of their inner workings.

## 3.1 Tilings

The kernels which produce a `HyperbolicTiling` currently available in the package are:

- `StaticRotationalSector` or `"SRS"`
  Cells are constructed via rotations about vertices of existing ones. Cells are implemented as `HyperPolygon` class objects and can be refined (compare Section 3.4). A bookkeeping system is used to avoid duplicate cells.

- `StaticRotationalGraph` or `"SRG"` (default)
  Algorithmically related to SRS, this kernel constructs adjacency relations between cells already during the construction of the tiling. It is currently the default tiling kernel of the package and provides methods for adding or removing cells from the lattice dynamically.

- `GenerativeReflection` or `"GR"`
  Very fast and lightweight tiling construction mechanism, which uses reflections on "open" edges to generate new cells. Only one symmetry sector is held on storage, with cells outside of this sector being generated on demand.

- `Dunham` or `"DUN07"`
  An implementation of the influential construction algorithm by D. Dunham [79, 93]. Recursive calls to a hierarchical tree structure are used to build duplicate free tilings in hyperboloid coordinates rather than in the Poincaré disk representation.

- `DunhamX` or `"DUN07X"`
  A modern, heavily optimized variant of `DUN07`, with a performance increase of more than one order of magnitude.

When instantiating a tiling, the kernel can be selected via a keyword argument, using either the abbreviation string

```
from hypertiling import HyperbolicTiling

T = HyperbolicTiling(7,3,2, kernel="GR")
```

or the full class name

```
from hypertiling import HyperbolicTiling
from hypertiling import TilingKernels

T = HyperbolicTiling(7,3,2,
      kernel=TilingKernels.GenerativeReflection)
```

An optional keyword for the kernels `"SRS"` and `"SRG"` is `center`, which can take on the values `cell` (default) and `vertex` and determines whether the tiling is centered around a polygon or a vertex. Examples for both cases can be found in Figure 1. In this context, it is worth to be remarked that a cell-centered $(p, q)$ tiling is the graphic-theoretical dual of a vertex centered $(q, p)$ tiling and vice versa.

An in-depth description of all kernels, including their particular advantages and shortcomings, as well as a detailed performance comparison can be found in Section 5. In most cases, however, the user is well served by either the `"SRG"` kernel (default), for greater flexibility, or the `"GR"` kernel, when speed is important or computing resources are a constraint.

Encapsulating all of the internal mechanics into kernel objects guarantees easy debugging, modification, and, most important, extendability of the package. Irrespective of which kernel has been selected for the construction, a `HyperbolicTiling` object provides, e. g. iterator functionality (which returns a list of coordinates of the cell center and vertices) and can return its

size via the Python built-in `len()` function. Moreover, cells come with several attributes, such as the coordinates of their vertices and center, angle in the complex plane, orientation, layer (or generation) in the tiling, and symmetry sector. These attributes can be accessed using get functions, for example

```
T.get_vertices(i)
T.get_angle(i)
T.get_layer(i)
```

which return these quantities for the $i$-th cell. A full reference of get-functions can be found in the package documentation. Note that the definition of polygon layers, the output of `get_layer`, might differ among kernels and can be unavailable for some (such as `"DUN07"` and `"DUN07X"`) due to algorithmic constraints.

## 3.2   Graphs

In addition to the kernels described in the previous section, we provide kernels which construct `HyperbolicGraph` objects. What is the difference between a `HyperbolicTiling` and a `HyperbolicGraph`? Broadly, in Section 3.1, cells are considered as individual entities, without knowledge of their surroundings. It is clear, however, that neighborhood relations are a crucial property in many applications, as detailed, e.g., in Section 1.2. For this reason, we provide methods for establishing adjacency between cells or, in other words, to access tilings as graphs. Such graphs capture the entire *geometric* structure of the tiling and can be represented, for instance, as an array of adjacent cell indices, similar to a sparse representation of the corresponding adjacency matrix. Kernels, which construct *only* this graph structure, independent of an actual representation of cells (in terms of their coordinates) are dedicated graph kernels and produce `HyperbolicGraph` objects. These leaner objects yield reduced features and functionality compared to a full `HyperbolicTiling`. Currently, two graph kernels are available in the package:

- `GenerativeReflectionGraph` or `"GRG"`
  Following the same algorithmic principles as the GR tiling kernel, this class constructs neighborhood relations already during the construction of the lattice. Only one symmetry sector is explicitly stored, whereas any information outside this sector is generated on demand. Geometric cell information, except for the center coordinates, is not stored.

- `GenerativeReflectionGraphStatic` or `"GRGS"`
  A static variant of the GRG kernel. Adjacency relations for all cells are explicitly computed, such that no sector construction and no on-demand generation is required. Hence the memory requirement is about a factor $p$ larger compared to GRG. Nonetheless, GRGS is still very fast and therefore particularly suited for large-scale simulations of systems with local interactions.

Compared to tilings, hyperbolic graphs are invoked via a separate factory pattern, shown in the following code example:

```
from hypertiling import HyperbolicGraph

G = HyperbolicGraph(7, 3, 2, kernel="GRG")
```

## 3.3   Neighbors

For kernels that do not establish adjacency relations already upon lattice construction, these relations can be computed in a separate second step, if required. The methods `get_nbrs` and

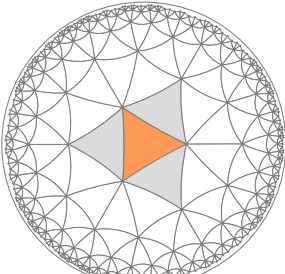 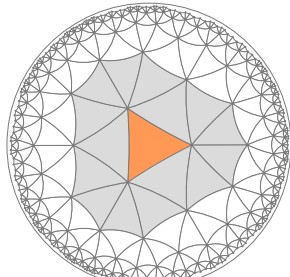 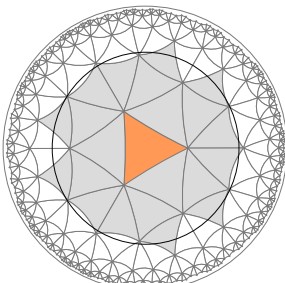

Figure 2: Adjacency can be defined by shared edges (left panel), shared vertices (middle panel) or within a radial region (right panel, black circle).

`get_nbrs_list`, internally implemented as wrapper functions, offer various neighbor search algorithms, selectable using the `method` keyword, as shown in the following code example:

```
# construct tiling
T = HyperbolicTiling(7, 3, 5, kernel="SRS")

# select algorithm "radius optimized slice" (ROS)
T.get_nbrs_list(method="ROS")
```

The default method is kernel specific and generally the fastest available. Output of `get_nbrs_list` is a sparse nested `List` of length $N$, where $N$ is the total number of polygons in the tiling. Sublist $i$ contains the indices of those cells which are neighbors of the cell with index $i$. The function `get_nbrs` is invoked with an explicit index and yields only the list of neighbors of that particular cell. Note that, even though `HyperbolicTiling` and `HyperbolicGraph` are different objects, neighbors are accessed in the same way

```
# let T be a HyperbolicTiling or HyperbolicGraph

# return neighbours of cell i
T.get_nbrs(i)

# return neighbours of all cells
T.get_nbrs_list()
```

For the SRS kernel, the available neighbor search methods include `method="ROS"` (radius optimized slice), where the distance between any pair of cells in one symmetry sector is computed and compared against the lattice spacing. Also, a combinatorial algorithm `method="EMO"` (edge map optimized), where adjacency relations are obtained by identifying corresponding edges among polygons, is provided. A detailed discussion of all neighbor methods of the SR kernel family can be found in Section 5.3.4. The GR kernel provides several different algorithms as well, including a radius search `method="radius"` and a geometrical algorithm exploiting the lattice construction mechanics, `method="geometrical"`. For a detailed list of all methods specific to the GR kernel, refer to Section 5.5.5.

Omitting the `method` keyword and hence resorting to the default algorithm is a solid choice in many use cases – unless an unusual definition of neighborhood is required. As illustrated in Figure 2, neighbors may for instance be defined by sharing either an edge or a vertex with the cell under consideration. But also broader neighborhoods are possible, e. g. by employing an appropriately tuned radius search. To ensure clarity regarding the specific definition employed by a particular `method`, refer to the package documentation.

In general, the computation of adjacency relations in an *existing* tiling presents a non-trivial task. Given the fact that a two-dimensional manifold of negative curvature can not be embedded into a higher dimensional Euclidean space, a natural ordering, which can be used to group or sort cells depending on their position, is lacking. Merely partitioning the Poincaré

disk into one or several rectangular grids, inspired by techniques like hierarchical multigrids, fails to achieve efficient ordering. In a loose sense, this issue arises due to the exponential growth of the manifold's volume in all directions. At the same time, the representation is heavily distorted towards the boundary as the unit circle is finite. As a result, neither the Poincaré disk coordinates nor any Euclidean-style grid can be employed for efficient sorting purposes.

Given these difficulties, a conceptually straightforward way of identifying adjacent cells in *any* regular geometry is by radius search, which is available as a standalone function in the `neighbors` module and used as the default algorithm behind `get_nbrs_list` in some kernels, such as DUN07. However, it should be emphasized that any neighbor search method that makes explicit use of Poincaré disk coordinates risks becoming inaccurate close to the unit circle, where the Euclidean distance between adjacent cells vanishes. For this reason, we recommend performing additional consistency checks whenever very large lattices are required.

### 3.4   Refinements

Unlike in traditional Euclidean structures, in a regular hyperbolic lattice, the edge length of a cell (which is equivalent to the effective lattice spacing) can not be tuned freely. This is a direct consequence of a non-zero curvature radius, which introduces an intrinsic length scale to the system. As discussed in detail in Section 4.3, shrinking or enlarging a regular polygonal cell also changes its shape. In particular, the interior angles at the vertices are altered, resulting in overlapping or voids in the tessellation. As a consequence, a continuum limit, where the lattice spacing tends to zero, is not trivially achievable.

To a certain extent, this limitation can be circumvented by introducing *refinements*. As demonstrated in Figure 3, in a triangular tiling, each triangle can always be subdivided into four smaller ones. The bisectors of the edges of the original triangle are used as new vertices. In principle, this process can be repeatedly applied and returns more and more fine-grained refinement levels. If the original tiling is not a triangular one, the first refinement step needs to be adjusted. In this case, we first subdivide every $p$-gon ($p > 3$) into $p$ uniform triangles, employing the center (of mass) of the original cell as a new vertex, shared among the new triangles. Note that these new cells can be highly non-equilateral, depending on the parameters $p$ and $q$. After this first refinement step, we can proceed as described above.

It is important to emphasize that triangles generated in the refinement procedure are no longer isometric. Even when starting from an equilateral triangle (such as in the first refinement step of a $(3, q)$ tiling), the central triangle will in general differ from the three outer ones. Also, neither of the four refined triangles is equilateral. Hence it is clear that the refined tiling is no longer maximally symmetric. Stated differently, hyperbolic lattice refinements always break the discrete rotational symmetry of the lattice as well as its strict regularity. This is the price we pay for denser tessellations. Whether or not it is an acceptable trade-off clearly depends on the application. One way to quantify the amount of non-uniformity in the refined lattice can be by monitoring the cell areas as refinement steps are added. The package provides convenient formulae to calculate angles, edges lengths and areas (see Section 4). As a general rule of thumb, the smaller the cells in the original, unrefined tiling are with respect to the radius of curvature, the less pronounced the resulting non-uniformity will be. By repeated application of refinements, the distribution of triangle areas will eventually saturate, as elementary triangles become increasingly flat on scales smaller than the curvature radius.

In summary, hyperbolic lattice refinements provide a clever way to circumvent some of the limitations that result from the peculiarities of hyperbolic geometry and they offer a continuum limit where the effective length scale in the lattice approaches zero. This comes at the cost of losing symmetry properties as well as strict uniformity of the lattice cells. Even though recent research demonstrates that for example bulk-to-bulk propagators on a refined lattice agree

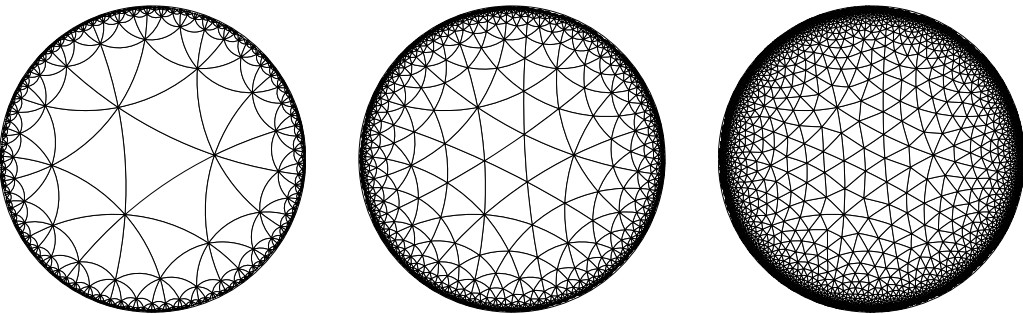

Figure 3: A cell-centered (3,8) lattice before (left), after one (middle) and after two (right) refinement iterations.

remarkably well with their continuum counterparts [70], these points should be kept in mind.

Currently, in the HYPERTILING package, refinements are supported in the SRS kernel. They can be invoked as follows

```python
from hypertiling import HyperbolicTiling

p, q, n = 6, 4, 3
T = HyperbolicTiling(p, q, n, kernel="SRS")

# add two refinement levels
T.refine(2)

# add three more
for i in range(3):
    T.refine()
```

The method `refine` takes the number of refinement levels as its only argument. One should be aware that the size of the tiling increases quickly with the number of refinement iterations. Specifically, the number of cells after $r$ refinement steps is given by

$$N_c(r) = \begin{cases} 4^r N_0 \\ 4^{r-1} p N_0 \end{cases} \quad \text{for} \quad \begin{matrix} p = 3 \\ p > 3 \end{matrix} \tag{6}$$

where $N_0$ represents the number of cells in the original, unrefined lattice.

## 3.5 Dynamic Modification and Filters

The static rotational graph (SRG) kernel comes with a number of unique features compared to other kernels in HYPERTILING, as it enables flexible modifications of an existing lattice. This can be useful, for instance, in applications where the lattice serves not only as a static supporting structure but also undergoes dynamic changes. An example might be reaction-diffusion processes where particles traverse the lattice in either a random or rule-based manner. In this scenario, it can be advantageous to dynamically generate the required lattice cells around the current positions of the particles, rather than generating and storing a vast, complete lattice. Moreover, a lattice that expands according to a walker's movement may avoid boundary effects.

An illustration of the SRG kernel's ability to add and remove cells dynamically is given in Figure 4. We start by adding cells to an existing tiling using the `add` method. This function acts on *exposed* cells, i.e. those with incomplete neighborhoods or, more precisely, cells that have less than $q$ neighbors. When invoked without arguments, all vacant spaces around *exposed* cells are filled with grid cells. Using an efficient container-based bookkeeping system (refer to

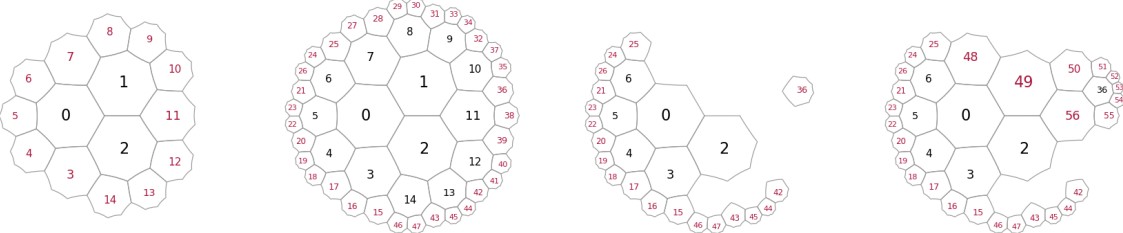

Figure 4: Demonstration of the lattice modification capabilities of the SRG kernel. Starting from a (7,3) lattice with two layers, an additional layer is added. Then, we remove a number of cells and arrive at a disconnected lattice, where we reconstruct new cells around cells 0 and 36. Exposed cells carry red labels.

Section 5.3.3 for an in-depth explanation) this can be accomplished without creating identical copies of already existing cells, so-called *duplicates*. Also, all absent cells which share a vertex with a currently exposed cell are created by `add`.

Considering that the SRG kernel gradually builds up the neighbor structure, it is important to note that exposed cells do only know about their "inwards" neighbors i. e. their parents, but not their siblings, until the next layer is generated. Naturally, cells located in the boundary layer of a tiling are always exposed.

When invoking `add`, also the associated adjacency relations are computed alongside the process of generating new cells. Newly created cells are always tagged as exposed, even in case they happen to close a gap or void within the lattice, as can be seen from Figure 4. Cells acted upon by `add` lose this attribute, as by design all possible neighbors are then present. In order to *un-expose* all cells, i. e. to obtain the complete graph structure of the current tiling, the user might consider calling a neighbor search method which globally computes the adjacency structure (see Section 5.3.4).

Apart from adding an entire new "layer", the `add` function can also be applied to a subset of cells, using a list of their indices as an input argument. Those can be exposed cells as well as "bulk" cells. Clearly, when a cell is already surrounded by the full set of $q$ neighbors, no new cells are created since any addition would be a duplicate. Newly added cells are assigned unique indices. The list of exposed cells can be queried using the `get_exposed()` class method.

The SRG kernel also provides the functionality to remove cells in an existing tiling, using the method `remove`, which takes a list of integers containing indices of cells, that are to be removed. Cells can be removed anywhere in the lattice, as demonstrated in Figure 4. As for the addition process, all local adjacency relations are updated accordingly.

In the SRG kernel, both the (usual) one-step lattice construction, as well as the dynamic addition of cells offer the option to incorporate *filters*. Filters are small helper functions, that can be implemented by the user. They can be used, for instance, to limit the generation of the tiling to certain regions in the Poincaré disk. Filters are expected to follow a simple syntax, namely they take a complex number (representing the central coordinate of a cell) as an input argument and return a boolean determining whether this cell is to be created or not. During the construction procedure, every candidate for a new cell is checked against that filter and only created if the associated condition is met. For example, in case one wants to grow a tiling only into the upper half of the complex plane, a filter like this might be suitable:

```python
import numpy as np

def my_angular_filter(z):
    angle = np.angle(z, deg=True)
    return True if (0 < angle < 180) else False
```

Concluding this section, it is worth noting that an interactive demonstration notebook is provided in the examples directory of the package. This notebook might serve as a resource to familiarize oneself with the aforementioned features and explore their functionality.

## 3.6 Drawing

Working with non-standard computational lattices also demands suitable methods for data visualization. In the Poincaré disk model of hyperbolic geometry, we are confronted with a number of challenges regarding graphical representations. First and foremost, lattice cells are always bound by curved edges, a property that is not natively supported by many standard plot engines. Moreover, for larger lattices, the strong distortion of the stereographic projection towards the boundary of the unit circle results in a pollution of cells near that boundary. This consumes substantial numerical resources, even though these cells typically can not be resolved.

To support the user to deal with those challenges, in HYPERTILING, we provide a selection of visualization routines for hyperbolic tilings and associated geometric objects. These routines are summarized in Table 1 and described in more detail below.

**Matplotlib API**

- `quick_plot`
  Focus on simplicity and speed, offering the best performance, but the fewest extra options, namely, adding the Poncaré disk boundary (unit circle), setting the image resolution and the arguments from matplotlib's `Polygon`, such as linewidth or alpha.

- `plot_tiling`
  Offers more customizable plots: by internally using matplotlib's `Patch`es instead of `Polygon`s, the full extent of matplotlib's keyword arguments is available. Besides, *lazy plotting* (see below) and individually colored cells are available.

- `plot_geodesic`
  Plots the tessellation with geodesic edges. The routines above plot polygons with Euclidean straight lines instead of hyperbolic geodesics, which is faster but of course not exact. In this routine, this limitation is partially overcome by converting edges to arc objects – the cost of this trick, however, is the loss of the notion of cells or patches, which therefore can not, e. g., be filled with color. We intend to include this feature in a future release.

| Feature | `quick_plot` | `plot_tiling` | `plot_geodesic` | `svg module` |
|---|---|---|---|---|
| performance | fast | medium | medium | slow |
| geodesic lines | ✗ | ✗ | ✓ | ✓ |
| individual cell color | ✗ | ✓ | ✗ | ✓ |
| lazy plotting | ✗ | ✓ | ✓ | ✗ |
| matplotlib kwargs | ✗ | ✓ | ✓ | ✗ |
| unit circle | ✓ | ✓ | ✓ | ✓ |
| backend | matplotlib | matplotlib | matplotlib | custom |

Table 1: Feature comparison of the current plotting capabilities of the HYPERTILING package. Here, *unit circle* denotes the option of adding the boundary of the Poincaré disk; *matplotlib kwargs*, the availability of matplotlib's keyword arguments; and *lazy plotting*, the possibility of setting a cutoff radius.

**Lazy plotting**   Since cells near the boundary of the unit circle often can not be displayed properly, it can be useful to set a cutoff radius beyond which polygons are omitted. This feature is activated by the option *lazy plotting*, available for the `plot_tiling` and `plot_geodesic` routines, and results in lighter plots without slicing the lattice itself.

**Code extension**   The modular design of the code simplifies the integration of specific features from the library into the user's custom routines. Two important internal plotting functions are given by `convert_polygons_to_patches` and `convert_edges_to_arcs`, which produce the corresponding matplotlib graphic objects from hyperbolic cells. The routines `plot_tiling` and `plot_geodesic` described above are essentially wrappers for these functions, which are useful building blocks for further plot scripts, as shown in our demo notebooks, available in the package repository.

**SVG module**

Many applications demand tessellations to be depicted in a geometrically accurate manner – e. g. with vertices connected by geodesics instead of Euclidean straight lines – and in a format that allows the usual graphical manipulations, such as filling areas with color. To make this possible we provide an SVG drawing extension module in HYPERTILING. The function `make_svg` renders the tiling as a vector graphic object where edges are drawn as geodesics and polygons are encoded as closed loops consisting of several edges. The SVG image can then be displayed by the interpreter using `draw_svg` or written to a file via `write_svg`.

   The SVG module provides in many respects the most flexible drawing option in the HYPER-TILING package since SVG images can be freely manipulated in any text editor or vector graphic program. The module's downside is its incompatibility with the matplotlib plot environment: The extension provides a few options, such as line color and thickness, however, production quality images typically demand additional polishing using external programs.

## 3.7   Animations

Besides its plotting facilities, HYPERTILING also comes with animation classes, which support dynamically adjustable viewpoints and cell colors. The resulting animations can be exported as video files. These classes are implemented as wrappers around matplotlib's built-in animation module and render particularly well in a Jupyter notebook environment, where interactive plotting can be activated using the `%matplotlib notebook` command. HYPERTILING's animation classes are briefly described below and a demonstration notebook with code examples is available in the package's repository.

**List Animations**   Given a pre-computed array-like object of color states, the `AnimatorList` creates an animation in which the cells' colors cycle through this list. Standard matplotlib animation keyword arguments are also accepted, including `interval`, setting the distance between color changes in milliseconds, or `repeat` or `repeat_delay`.

**Live Animations**   The `AnimatorLive` class has a very similar signature to that of `AnimatorList`, but instead of taking a pre-computed list of states, it allows the color values to be updated dynamically, according to a user-implemented function. This function must take the current state as its first argument and return a new state (an array-like object matching the number of cells in the tiling). Additional function arguments (such as physical parameters) can be passed as keyword arguments using `stepargs`. As with `AnimatorList`, matplotlib animation keyword arguments are passed using the `animargs` dictionary. Note that the argument `frames` in this case controls the duration of exported animations.

**Path Animations**   The classes above vary only the colors on an otherwise static background tiling. The `PathAnimation` class introduces the possibility of translating the lattice during the animation. Colors can also be animated, according to a list, like in `ListAnimation`. The translation is defined by a path, which can be a sequence of polygon indices or of coordinates in the complex plane: These path elements are sequentially moved to the origin, with the smoothness of the animation being controlled by the number of intermediate frames, `path_frames`, between every path element.

## 4   Mathematical Foundations

We now transition to the mathematical foundations of two-dimensional hyperbolic geometry. This section explains key concepts such as isometries, Möbius transformations, and the construction of geodesics and polygons in the Poincaré representation.

### 4.1   Isometries

As already hinted in the introduction, the hyperbolic 2-space in the Poincaré disk representation is given by the interior region of the complex unit circle, $\mathbb{D}_2 = \{z \in \mathbb{C}, |z| < 1\}$, equipped with the metric

$$\mathrm{d}s^2 = \frac{4\ell^2}{(1 - z\bar{z})^2} \mathrm{d}z \mathrm{d}\bar{z}, \tag{7}$$

where $K = -1/\ell^2$ represents the constant negative Gaussian curvature of the manifold. The set of orientation-preserving Möbius transformations $f : \mathbb{C} \to \mathbb{C}$, defined as

$$f(z) = \frac{az + b}{cz + d}; \quad a, b, c, d \in \mathbb{C}; \quad ad - bc \neq 0 \tag{8}$$

establishes the Möbius group

$$\mathrm{M\ddot{o}b}_+ \cong \mathrm{PGL}(2, \mathbb{C}), \tag{9}$$

which is isomorphic to the projective linear group of degree two with complex coefficients. These transformations take on a key role in hyperbolic geometry since certain subgroups act as conformal isometries on the Poincaré disk $\mathbb{D}_2$. Specifically, the subgroup of Möbius transformations which describes all orientation-preserving isometries of $\mathbb{D}_2$ and contains all elements as in Eq. (8) can be written as $2 \times 2$ matrices

$$\begin{pmatrix} z \\ 1 \end{pmatrix} \mapsto \begin{pmatrix} a & b \\ c & d \end{pmatrix} \begin{pmatrix} z \\ 1 \end{pmatrix}. \tag{10}$$

Through the unitarity requirement, we find $|a|^2 - |b|^2 = 1$ as well as $d = \bar{a}$ and $c = \bar{b}$, yielding the final form

$$\begin{pmatrix} z \\ 1 \end{pmatrix} \mapsto \underbrace{\begin{pmatrix} a & b \\ \bar{b} & \bar{a} \end{pmatrix}}_{M} \begin{pmatrix} z \\ 1 \end{pmatrix}, \tag{11}$$

with $|M| = 1$. This makes it manifest that the set of orientation-preserving isometries forms a projective special unitary group

$$\mathrm{PSU}(1, 1) = \mathrm{SU}(1, 1) / \{\pm \mathbb{1}\} \subset \mathrm{M\ddot{o}b}_+, \tag{12}$$

which is isomorphic to PSL$(2, \mathbb{R})$ [39]. Discrete subgroups of PSU$(1, 1)$ are oftentimes referred to as the *Fuchsian groups* in the literature [38].

Depending on the choice of the three independent coefficients, $M$ represents elementary isometric transformations of the hyperbolic plane such as rotation, translation, and reflections. Factoring a complex angle, we arrive at

$$f(z) = e^{i\phi} \frac{z - \alpha}{1 - \bar{\alpha} z}, \quad 0 \leq \phi < 2\pi, \quad \alpha \in \mathbb{D}_2 \tag{13}$$

and setting $\alpha = 0$ we find rotations about the origin $z \mapsto e^{i\phi} z$, which in matrix form can be written as

$$R(\phi) = \begin{pmatrix} e^{i\phi/2} & 0 \\ 0 & e^{-i\phi/2} \end{pmatrix}. \tag{14}$$

Translations on $\mathbb{D}_2$ are proper Lorentz boosts in $2 + 1$ dimensions, or in other words elements of SO$^+(1, 2)$. Consequently, we may write translations as

$$\begin{pmatrix} z \\ 1 \end{pmatrix} \mapsto \underbrace{\begin{pmatrix} \cosh \frac{\theta}{2} & \sinh \frac{\theta}{2} \\ \sinh \frac{\theta}{2} & \cosh \frac{\theta}{2} \end{pmatrix}}_{T_x(\theta)} \begin{pmatrix} z \\ 1 \end{pmatrix} \tag{15}$$

for boosts along the real axis with rapidity $\theta$ (not $\theta/2$!). Boosts in an arbitrary direction can readily be realized by adding suitable rotations before and after the actual translation, i.e.

$$T(\theta) = R(\phi) T(\theta) R(-\phi), \tag{16}$$

where $\phi$ is the angle between a vector pointing in the translation direction and the positive $x$-axis. Likewise, a rotation around an arbitrary point can be accomplished through the proper composition of elementary transformations. Thus, in addition to rotation, we implement the general form of a translation of point $z_0$ to the origin, given by

$$z \mapsto f_T(z) = \frac{z - z_0}{1 - z \bar{z}_0} \tag{17}$$

where the inverse transformation is given by

$$z \mapsto f_{T^{-1}}(z) = \frac{z + z_0}{1 + z \bar{z}_0} \tag{18}$$

which maps the origin back to $z_0$.

## 4.2 Geodesics

On the Poincaré disk, straight lines are circular arcs which intersect orthogonally with the boundary, i.e. the unit circle $\mathbb{S}^1$, as illustrated in Figure 5. Points on $\mathbb{S}^1$ (compare $u, v$ in the Figure) are located infinitely far away from the interior region of the disk in terms of geodesic distance. Geometrically, the construction of a geodesic line through two points $z_a, z_b \in \mathbb{D}_2$ requires a circle inversion on the set of complex numbers without the origin, i.e. $f : \mathbb{C}^+ \to \mathbb{C}^+$, where $f(z) = 1/\bar{z} = z/|z|^2$ of either of the two points, which gives us $z_c$ outside the unit circle. We are left to construct a circle through the points $z_a$, $z_b$, $z_c$ and suitably parametrize the segment from $z_a$ to $z_b$. Note that the circle inversion can not be applied if $z_a$, $z_b$ and the origin are collinear. In this case, the geodesic becomes a straight line in the projection.

One particular advantage of the Poincaré disk representation of two-dimensional hyperbolic space is that it represents a conformal model. Therefore, angles on $\mathbb{D}_2$ are measured

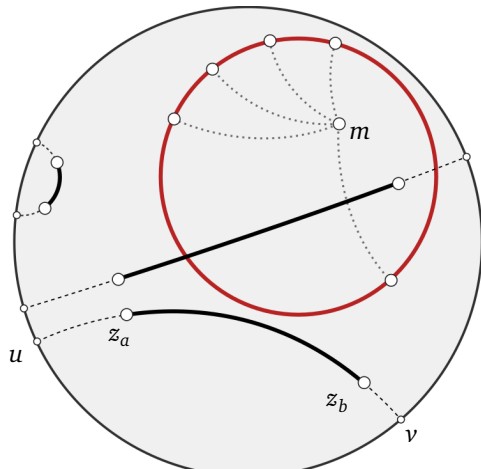

Figure 5: Geodesic line segments (black arcs) and their continuation towards infinity (dashed lines). The boundary of the gray disk represents the complex unit circle. The point $m$ denotes the center of a hyperbolic circle and has equal distance to all points on its boundary (red). Corresponding shortest paths are marked by dotted lines.

exactly as the corresponding Euclidean angle in $\mathbb{C}$ and hyperbolic circles are mapped to Euclidean circles, although hyperbolic and Euclidean circle centers and radii do not coincide, as illustrated in Figure 5.

We define the length of a parametrized path $\gamma : [a, b] \to \mathbb{D}_2$ as a suitable integral over the hyperbolic line element

$$\text{length}_{\mathbb{D}_2}(\gamma) = \int_a^b \left\| \frac{d\gamma}{dt} \right\|_{\mathbb{D}_2} dt \equiv \int_a^b \frac{2\ell}{1 - |\gamma(t)|^2} \left\| \gamma'(t) \right\|_2 dt \tag{19}$$

$$= \int_\gamma \frac{2\ell}{1 - |z|^2} |dz| \tag{20}$$

where $\|\cdot\|_2$ represents the Euclidean norm. This allows to compute geodesic distances between two points $z_a$, $z_b$ on the Poincaré disk by evaluating the above integral along the corresponding shortest path

$$d(z_a, z_b) = \inf \left\{ \text{length}_{\mathbb{D}_2}(\gamma) : \gamma \text{ with endpoints } z_a, z_b \right\}. \tag{21}$$

The hyperbolic distance between a point $z \in \mathbb{D}_2$ and the origin yields

$$d(0, z) = 2\ell \tanh^{-1}(|z|) = \ln\left( \frac{1 + |z|}{1 - |z|} \right), \tag{22}$$

whereas the general form of the distance between two points $w, z \in \mathbb{D}_2$ is given by

$$d(w, z) = 2\ell \tanh^{-1}\left( \frac{|z - w|}{|1 - z\bar{w}|} \right). \tag{23}$$

## 4.3 Polygons

We define a hyperbolic *polygon* as a region confined by a set of geodesic line segments, so-called *edges*. The endpoints of these edges are called *vertices*. If all edges have the same length, the polygon is said to be regular. Unlike in Euclidean geometry, where the sum of the inner angles

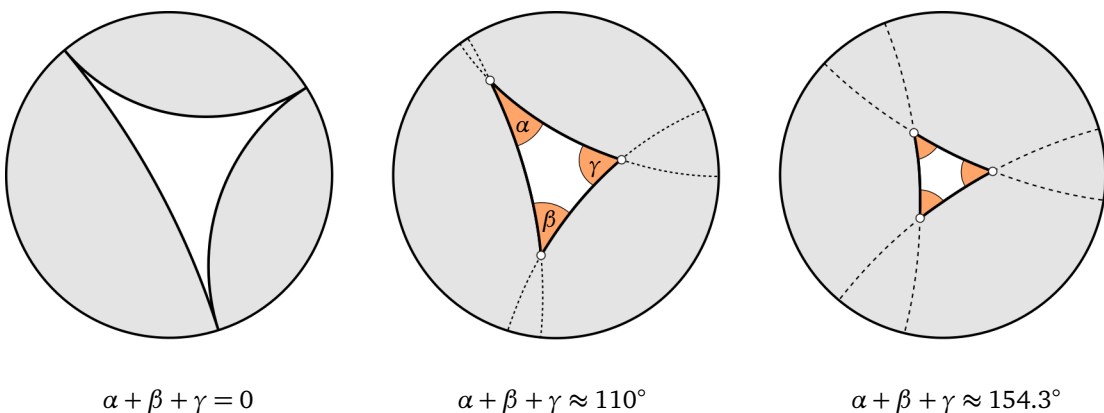

$$\alpha + \beta + \gamma = 0 \qquad\qquad \alpha + \beta + \gamma \approx 110° \qquad\qquad \alpha + \beta + \gamma \approx 154.3°$$

Figure 6: Hyperbolic triangles of different size. The left panel depicts a so-called *ideal triangle*, where all three vertices are located at infinity (ideal points) and the sum of interior angles is zero. The middle panel shows a finite triangle and the right panel the fundamental cell of a regular (7,3) tiling, with equal edge lengths and an angle sum of exactly $2\pi p/q$.

at the vertices is exactly $(p-2)\pi$, polygons in hyperbolic (spherical) spaces have angle sums of less (more) than $(p-2)\pi$, respectively.

A *tiling* or tessellation is a set of regular polygons which are isometric and cover the entire manifold without overlappings or voids. In the Euclidean case there exist exactly three isometric tilings, which are the well-known square, triangular and honeycomb lattices [94]. The corresponding characteristic lattice spacing $h$ can be scaled freely. We encounter a substantially different behavior in a hyperbolic space, where the curvature radius introduces an additional length scale. As a result, the sum of inner angles depends on the size of a polygon. In Figure 6 we demonstrate the interplay between polygon areas and its vertex angles. From the picture, it becomes clear that as the triangle size decreases, the angle sum approaches $\pi$. This can be intuitively understood as the space becoming more and more flat locally. Additionally, at any point in the manifold, the local circumangle must sum up to $2\pi$. These constraints result in the fact that the fundamental (and any other) polygon must possess specific dimensions to enable the tiling to cover the entire disk. Specifically, in order to accomplish a full covering, the angles at the vertices need to have an angle of exactly $2\pi/q$, as shown, for instance, in the rightmost panel of Figure 6. Here, angles are given by $\alpha = \beta = \gamma = 2\pi/7$.

Stated differently, the shape of cells in a regular tiling depends on the Schläfli parameters $p$ and $q$. In Figure 7 we show a more detailed illustration of all fundamental angles in a regular tesselation. In this example of a (6,4) lattice, the size of the hexagonal cells is determined by the restriction that the angle at the edges is exactly $2\beta = 2\gamma = 2\pi/q = 90°$. A larger size (and consequently smaller inner vertex angles) would result in gaps between adjacent hexagon cells along their edges. A smaller size (and consequently larger angle) would result in overlaps.

The fact that the choice of $p$ and $q$ fixes the polygon size, is a crucial property to keep in mind when working in hyperbolic geometry. It means that the characteristic length of the system can not be tuned. We list the most important geometric lengths in Table 2 for a selection of $p$ and $q$ values. Functions that help to compute these quantities are provided in the `hypertiling.util` module. Their exact definition is discussed in the following:

The circumradius $r_0$ of a polygon, which is the distance from its center to either of its

vertices is given by

$$r_0 = \sqrt{\frac{\cos\left(\frac{\pi}{p} + \frac{\pi}{q}\right)}{\cos\left(\frac{\pi}{p} - \frac{\pi}{q}\right)}}. \tag{24}$$

Furthermore, the radius $r_m$ of the in-circle, i.e. the largest circle centered at the origin and fully inside the polygon is implicitly given by the expression

$$\cos\left(\frac{\pi}{p}\right) = \frac{\tanh\left(2\tanh^{-1} r_m\right)}{\tanh\left(2\tanh^{-1} r_0\right)}. \tag{25}$$

Recall that $r_0$ and $r_m$ are distances in the complex plane. The true geodesic distance can easily be computed via Equation (22) as $d(0, r_0)$ and $d(0, r_m)$, respectively.

For computations involving triangles, hyperbolic geometry provides a particularly useful rule, the so-called law of sines. Given a generic hyperbolic triangle, where $a$, $b$ and $c$ represent the edge lengths opposite to the respective vertices $A$, $B$, $C$, one finds

$$\frac{\sin\alpha}{\sinh(a)} = \frac{\sin\beta}{\sinh(b)} = \frac{\sin\gamma}{\sinh(c)}. \tag{26}$$

It is always possible to divide the fundamental polygon into $2p$ isometric triangles (compare $\triangle ACD$ in Figure 7). In this case, where the inner vertex angle at $D$ is exactly $\pi/2$, the relation reduces to

$$\sin\frac{\alpha}{2} = \frac{\sinh\overline{DC}}{\sinh\overline{AC}} \tag{27}$$

which is equivalent to

$$\sin\frac{\pi}{p} = \frac{\sinh(h/2)}{\sinh d_0}. \tag{28}$$

The letter $h$ denotes the hyperbolic distance between vertices, which is nothing but the effective lattice spacing and can be computed as

$$h = h^{(p,q)} = 2\ell\cosh^{-1}\left(\frac{\cos\left(\frac{\pi}{p}\right)}{\sin\left(\frac{\pi}{q}\right)}\right). \tag{29}$$

| $\{p, q\}$ | $h^{(p,q)}$ | $h^{(q,p)}$ | $h_r$ | $r_0$ |
|---|---|---|---|---|
| $\{3, 7\}$ | 0.566256 | 1.090550 | 0.620672 | 0.300743 |
| $\{4, 5\}$ | 1.061275 | 1.253739 | 0.842481 | 0.397975 |
| $\{5, 4\}$ | 1.253739 | 1.061275 | 0.842481 | 0.397975 |
| $\{6, 4\}$ | 1.762747 | 1.316958 | 1.146216 | 0.517638 |
| $\{7, 3\}$ | 1.090550 | 0.566256 | 0.620673 | 0.300743 |
| $\{8, 4\}$ | 2.448452 | 1.528571 | 1.528570 | 0.643594 |
| $\{12, 10\}$ | 3.951080 | 3.612418 | 3.132385 | 0.916418 |

Table 2: Characteristic geometric lengths for a selection of different regular tilings, including the lattice spacing $h^{(p,q)}$ (i.e. the edge length), the lattice spacing of the dual lattice $h^{(q,p)}$ and the cell radius $h_r$ (i.e. the geodesic distance between the cell midpoint and its vertices), all three in units of $\ell$. The last column displays the radius of the fundamental cell in the Poincare disk, given by $r_0 \in (0, 1)$.

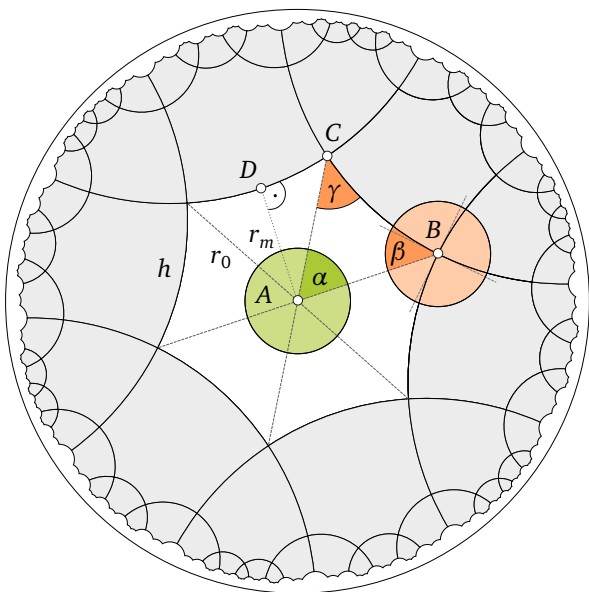

Figure 7: Example of the fundamental polygon (white), its subdivision into isometric triangles and associated angles in a (6,4) tiling. Angles are given by $\alpha = 2\pi/p$ and $\beta = \gamma = \pi/q$. Moreover $h = \overline{BC} = \overline{DC}/2$ represents the lattice spacing, i.e. the geodesic distance between any two adjacent vertices in the tiling.

Finally, the area enclosed by a general triangle in the hyperbolic domain is given by

$$A_\triangle = (\pi - \alpha - \beta - \gamma)\ell^2 \tag{30}$$

and allows straightforward computation of the polygon area according to Figure 7.

# 5 Architecture

## 5.1 Codemap

The HYPERTILING library offers a number of different methods for constructing hyperbolic tilings and graphs. At the heart of the package, these are implemented as *kernels*, each of which contains its own construction algorithm, memory design, auxiliary functions and specific manipulation features.

This section aims to provide an overview of the relationships among the parts of the library. These are summarized in the hierarchy of classes and modules depicted in Figure 8. Both hyperbolic tilings and graphs (see Sections 3.1 and 3.2) are typically instantiated through factory functions, highlighted in orange in the diagram. Alternatively, objects can be directly created using constructors of the respective kernel classes. All kernel classes are required to implement the respective abstract base class, either for graph or tiling, marked in yellow. The actual kernel classes themselves are represented in purple. Some kernels share common foundational structures and inherit from shared base classes, depicted in blue. This streamlines the development and maintenance of the library by promoting code reusability. Utility classes, which are essential for the functionality of kernels, but are separated into distinct classes, are also color-coded blue in Figure 8.

A range of modules, indicated in white in the diagram, provide auxiliary functions such as distance calculations, coordinate transformations and more. These modules are universally

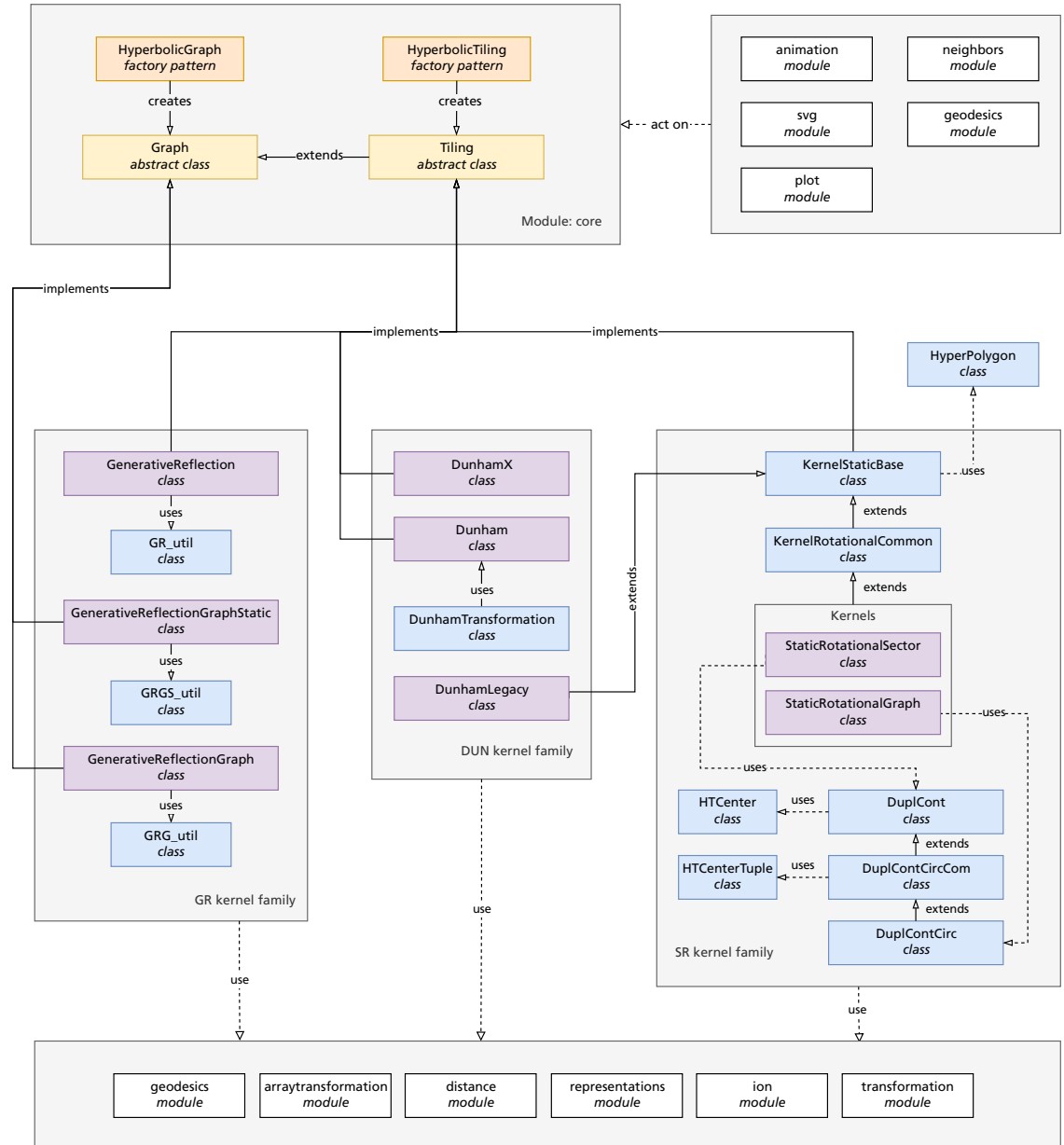

Figure 8: UML type diagram for the HYPERTILING library. Rectangles represent classes or modules, while connections between them describe their relationships. Color code: Factory function: orange; abstract base class: yellow; kernel class: purple; shared base class: blue; auxiliary and extension modules: white. Gray regions denote different functional families.

accessible to all kernels and utility classes due to the common interface provided by the abstract base classes. Additional modules for plotting, animation, and neighbor calculations further extend the capabilities of constructed tilings and graphs. These modules typically act upon the tilings and graphs to provide, e.g. visual representations.

Due to their extensive number, the figure omits specific class methods. For a comprehensive overview of available functions and detailed API usage, the package documentation can be consulted.

## 5.2   Terminology

A property that is shared among all construction algorithms is that they work incrementally, with new polygons being generated from existing ones. Hence before we start to dive deeper into kernel-specific properties, it is useful to define commonly used family relations, relative to a given polygon:

- *self*: The polygon under consideration itself

- *parent*: The polygon from which *self* is created

- *sibling*: A polygon also created by *parent*, but which is not *self*

- *child*: A polygon created by *self*

- *nibling*: A polygon created by a sibling

In what follows, the internal mechanics of all kernels currently available in the package is discussed in greater detail. We remark that this discussion is quite technical. If the reader does not require this level of detail at present, they might choose to skip Sections 5.3 – 5.5 and refer to it on an as-needed basis later.

## 5.3   Static Rotational Kernels

The family of static rotational kernels comprises two distinct implementations, namely the *static rotational graph* (SRG) kernel, as well as *static rotational sector* (SRS) kernel. From the perspective of features, we have already discussed the unique lattice manipulation capabilities and immediate graph generation of SRG, as well as the ability to refine lattices in SRS, which moreover uses an optimized sector construction, in Section 3.1. In this chapter, we shed light on the internal algorithmic design and the data structures used in these kernels. In order to better understand the program workflow, we also include a UML-like diagram in Figure 9, which provides a visual representation of the relationships between classes within the SR kernel family.

### 5.3.1   General Idea

Both the SRG and SRS kernel are built upon a common, rather simple principle: New cells (internally stored as `HyperPolygon` objects in an array structure) are generated via rotations about the vertices of existing ones. Building a lattice starts with the construction of a fundamental cell, whose edge length is determined by geometric properties of the hyperbolic space (compare Section 4.3). In the case of a cell-centered tiling (which is the default option, and can be manually set by adding the `center="cell"` keyword in the `HyperbolicTiling` factory function) this fundamental polygon represents the first layer of the tessellation, whereas for `center="vertex"` the innermost layer consists of $q$ polygons (compare Figure 1). Next, the second layer is constructed by iterating over each vertex in the first layer and computing all adjacent polygons using successive rotations by an angle of $2\pi/q$ about that vertex. Technically, this operation can be split into a sequence of Möbius transformations, as detailed in Section 4.1. First, a translation is carried out, which moves the vertex to the origin; there, the polygon is rotated by $2\pi/q$ and the inverse translation brings the vertex back to its original position.

It is evident that the construction scheme presented above produces *duplicates*. For instance, adjacent parents create a number of identical children. As we aim to produce a proper duplicate-free tiling, a mechanism that avoids these identical copies is required. This

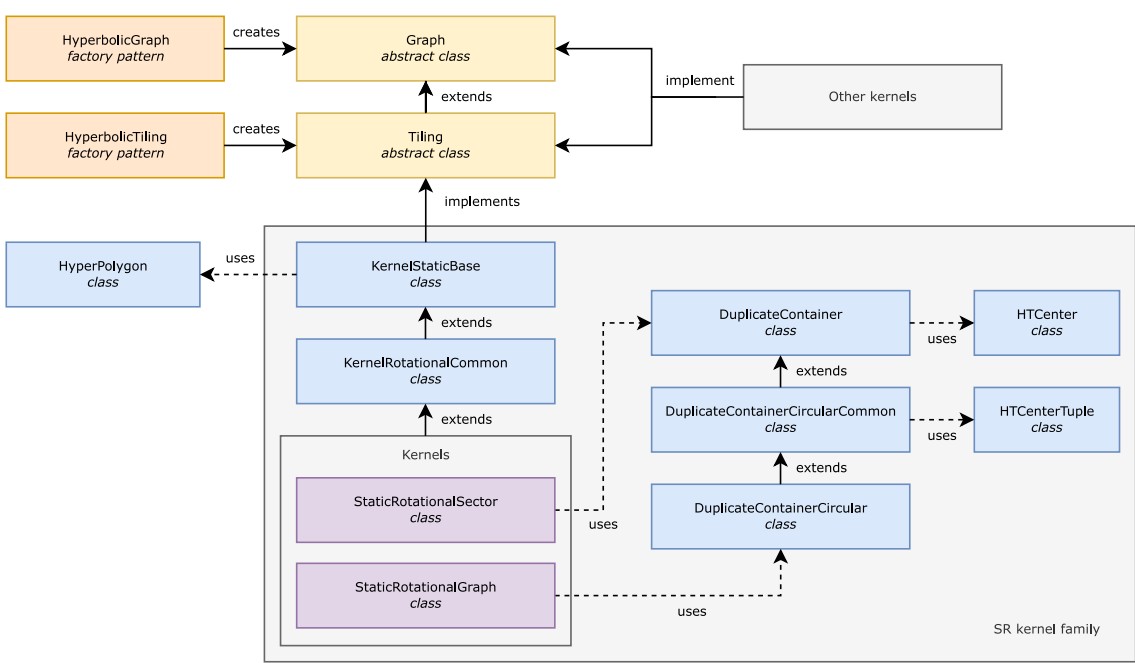

Figure 9: Class and program flow diagram for the SR kernel section of the package.

is achieved by introducing an auxiliary data structure we call `DuplicateContainer`. In this container, the coordinates of the center of every cell already present in the computed tiling are stored. Newly constructed cells are compared against existing ones and those already present in the container will be discarded.

### 5.3.2 Rotational Duplicates

As the name already indicates, the static rotational sector (SRS) kernel takes advantage of the $p$-fold rotational symmetry of a hyperbolic $(p,q)$ tiling. Only one sector of the lattice is explicitly constructed. In a second step, this fundamental sector is replicated $p$ times, accompanied by a suitable shift of attributes (such as vertex coordinates) to obtain the full lattice. As only one sector of the tiling is constructed, newly generated polygons, whose center coordinates end up being located outside the angle interval $0 \leq \phi < 2\pi/q$ are immediately discarded[4].

However, due to the overall high degree of symmetry in the tiling, it can occur that a new cell is positioned exactly on a symmetry axis, with its center numerically close to one of the sector boundaries. As a consequence, so-called *rotational duplicates* might be created. This issue can be illustrated using a specific example, as in Figure 10, where the boundaries between sectors are visualized by the dotted black lines. In the picture, the central cell constitutes layer 1 and the cell with index 1 is located in layer 2. Moving outwards, we recognize that both, polygons 2 and 14, are positioned directly at opposite sector boundaries and that they are identical in terms of a $p$-fold discrete rotational lattice symmetry. Hence only one of them belongs into the fundamental sector. We always decide to adopt the one with the smaller angle in the complex plane, hence, in this case, polygon 2. In practice, however, due to the closeness of both polygons to their associated sector boundary, combined with the finite numerical precision, four cases can occur:

1. Both polygons end up in the sector

2. Neither polygon ends up in the sector

---

[4]For vertex-centered tilings $2\pi/q$ is to be replaced by $2\pi/p$ here and in the following.

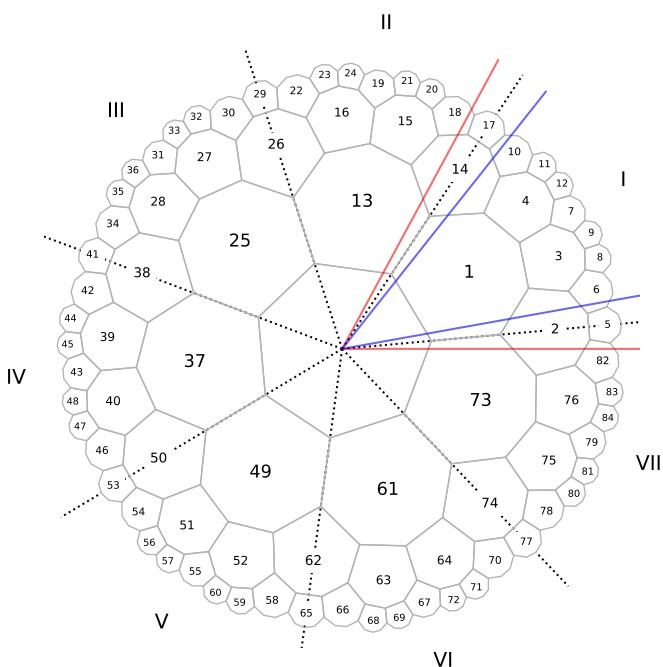

Figure 10: A hyperbolic lattice, divided into the fundamental symmetry sectors, separated by dotted lines and labeled by Roman numerals. Colored straight lines indicate the *soft* sector boundary of $\pm\epsilon$, which is used to filter out rotational duplicate cells as described in the text.

3. Only the one with the *smaller* angle ends up in the sector

4. Only the one with the *greater* angle ends up in the sector

In the first case, we end up with one *rotational duplicate*. Note that, in this example, polygons with indices 2 and 14 are no duplicates per se, i.e. considering only the sector they would not be recognized as duplicates due to their different center coordinates. However, upon the angular replication step polygon 2 will be rotated by $2\pi/p$ around the origin and then exactly coincide with polygon 14.

It is obvious that scenarios 1 and 2 must be avoided and we select scenario 4 to be the desired one. As a result of what is discussed above, a mechanism that reliably filters out potential rotational duplicates is required. To this end, we introduce a second instance of the `DuplicateContainer` data structure. Similar to the global duplicate management container, this second container stores the cells' center positions during the construction, however only those being located within a tiny slice around the lower sector boundary, i.e. with angles $|\phi| < \epsilon$, where $\epsilon \ll 1°$. Including these cells automatically ensures that options 2 and 3 can not be realized, since the polygon with the smaller angle is always inside this "soft boundary". After the entire sector is generated and before the angular replication of the lattice is carried out, the auxiliary container can be used to implement a filtering step. We loop over all cells in the soft boundary around the *upper* sector border, i.e. those with an angle of $2\pi/p-\epsilon < \phi < 2\pi/p+\epsilon$ and perform a clockwise rotation by $2\pi/p$. In Figure 10 this soft boundary region is indicated by the colored lines. If polygon 14 had accidentally been included into the fundamental sector, after this rotation its center position would be identical to that of polygon 2, which is already inside the large duplicate container. As a consequence, polygon 14 is deleted from the tiling, since it will be generated as a copy of polygon 2 at the replication stage of the algorithm.

### 5.3.3 Implementation Details

As discussed above, one of the key ingredients of any kernel in the SR family is the reliable detection of duplicate cells in the lattice. For the implementation of this `DuplicateContainer`, one might naively consider Python's `set` data container, since due to its property as a hashed data type, it allows find operations in constant to linear time complexity, depending on the number of elements. Nonetheless, due to the representation of cell center coordinates as floating-point numbers, it becomes necessary to round them to a specific number of decimal places. This rounding process is necessary to enable the hashing of coordinates but effectively leads to the creation of bins in the set. Now, for very large tilings, where cells will be located increasingly close to the unit circle, it might occur that two separate (i. e. non-duplicate) polygons are not readily distinguishable and could be mistakenly disregarded. Furthermore, and even more severe, the binning mechanism always has a chance to accidentally miss duplicates, if both coordinates are rounded to adjacent bins. Due to the lattice being constructed in an iterative fashion, minor computational inaccuracies might accumulate when proceeding outwards. This way, a more precise rounding could paradoxically lead to an increased occurrence of errors, since more bin boundaries are present.

In order to avoid binning issues altogether, we introduce a robust mechanism for managing duplicates. This is accomplished by a specialized data structure, the `DuplicateContainer`. When available on the system, we utilize the external `sortedcontainer` library, which enables searching within the data structure in logarithmic time $\mathcal{O}(\log(N))$, where $N$ denotes the number of cells already constructed. If it is not available, then our own, slower, fallback implementation is used. Since the centers of a polygon are complex numbers we first need to establish an ordering relation in order to utilize a binary tree-like data structure. In practice, we employ the complex *angles* of the center coordinates for this purpose, hence allowing for efficient access into a sorted container type. As a result, binning can be avoided altogether, and we are able to efficiently decide whether a polygon has already been constructed, with an accuracy up to nearly the relative machine precision. As a double check, during the lattice construction, the typical Euclidean distance of adjacent cells in the complex plane is monitored and a warning is issued in case this distance comes close to machine precision. This sets a theoretical limit on the available size of lattices in terms of the radial distance from the origin.

Additionally to this "upper" error limit, we also need to account for the accumulation of errors due to repeated calculation of layers. In order to keep track of this type of uncertainty we take a sample of polygons from every newly constructed layer and compute the distances to their corresponding neighbors. In particular, we are interested in the *minimal geodesic* distance within the sample, since this quantity can be interpreted as an approximate bound of the accumulated error, given the sample is large enough. We compare this minimal distance with the fundamental geodesic distance of the tiling (i. e. the lattice spacing, compare Section 4.3). In a scenario of error-free arithmetic, both values would be identical. However, due to the accumulation of gradually applied Möbius transformations, this discrepancy does not vanish but usually increases towards the outer layers. Once it reaches the order of magnitude the coordinates are rounded to (our binning size), the construction becomes unreliable and the user is duly warned.

### 5.3.4 Neighbors

When it comes to adjacency (or neighbor relations) between polygons, SRG and SRS follow fundamentally different approaches. The static rotational graph (SRG) kernel is capable of constructing the local neighborhood of cells already during the generation of the tiling. This is accomplished by a suitable generalization of the `DuplicateContainer` class (compare Figure 9) which can hold tuples consisting of center coordinates and corresponding cell indices.

This allows to not only detect whether a candidate polygon is already in the lattice (compare discussion above) but also to return its index in case it is already present. Since according to the SR construction principle *all* adjacent cells of the polygon under consideration (*self*) are attempted to be created, this way we can recover mutual neighbor relations; either the candidate polygon already exists, in which case its associated index is added to the neighbor list of *self*– or the candidate is, in fact, a valid new polygon, in which case it is assigned a unique new index and *this* index is being added to the list of neighbors. Clearly, in both cases, the index of *self* is added to other polygon's neighbor list, to establish mutual adjacency.

The generalized duplicate container furthermore opens the opportunity to dynamically remove cells, as has been demonstrated in Section 3.5. In principle, this would be possible with the standard duplicate container as well, but the requirement to identify cells based on floating point comparisons of their center coordinates would make this approach prone to rounding errors. Instead, having a cell index available in the container allows to remove cells by their indices, and use the center coordinate as a helper structure to quickly pinpoint the corresponding position in the array. In summary, removing a polygon *self* from the tiling requires the following steps:

1. Remove corresponding `HyperPolygon` from list of polygons

2. Remove neighbor list of *self* from neighbor array

3. Remove occurences of *self*'s index in neighbor lists of other polygons

4. Remove corresponding entry from duplicate container

5. Remove index of *self* from list of exposed cells

In order to perform these operations efficiently, it is necessary that the list of polygons and the list of neighbors in the SRG kernel are implemented as Python dictionaries. Only this way, entries can be added and removed by index in constant time. Overall, step 4 presents the bottleneck of the removal operation since, as detailed above, locating the correct entry in the duplicate container, in general, requires logarithmic time (binary search in a sorted container).

Turning to the SRS kernel, where neighborhood relations are not computed upon construction, we provide several different methods to obtain them in a separate step. All methods can be accessed by using the standard wrapper function `get_nbrs_list` via the parameter `method`, which can take on the following values:

- `get_nbrs_radius_brute_force` or `method="RBF"`
  If the distance between any pair of polygons falls below a certain threshold, they are declared neighbors. The radius can be passed as a keyword argument. In case no radius is provided, the lattice spacing is used. Note that this algorithm is very reliable but too slow for large tilings due to the brute-force all-to-all comparison.

- `get_nbrs_radius_optimized` or `method="RO"`
  Similar to `"RBF"` but uses numpy to gain a significant performance improvement.

- `get_nbrs_radius_optimized_slice` or `method="ROS"`
  Variant of the `"RO"` scheme, which exploits the discrete rotational symmetry and applies the radius search only to a $p$-fold sector of the tiling. Neighbors outside the fundamental sector are obtained via suitable index shifts.

- `get_nbrs_edge_map_optimized` or `method="EMO"`
  Locating neighbors by identification of shared edges. Apart from the initialization of the edges, this is a coordinate-free, combinatorial algorithm.

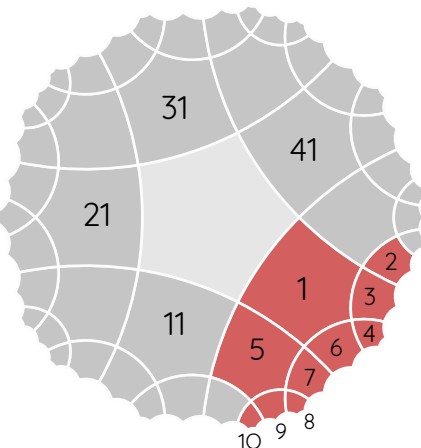

Figure 11: Sequence in which cells are created in the Dunham algorithm. First layer polygons {1,11,21,31,41} can be seen as initial nodes of independent branches, resembling the concept of a depth-first search. To enhance clarity, numerical labels are displayed only in the first branch.

- `get_nbrs_edge_map_brute_force` or `method="EMBF"`
  Brute-force variant of `"EMO"`, available for testing and debugging purposes.

## 5.4 Dunham's Algorithm

An important milestone in the development of algorithms for the construction of hyperbolic tilings is the combinatorial method presented by D. Dunham and coworkers in a series of publications in the early 1980s [79, 95]. A hierarchical spatial tree structure is employed, where recursively new trees of cells are created at the vertices of existing ones, resembling a depth-first search algorithm. Similarly to our static rotational kernels (Section 5.3), new polygons are created locally by discrete rotations of existing ones around their vertices. The first polygon in each tree creates one child less, as can be seen in Figure 11. The algorithm uses hyperboloid (also called Weierstrass) coordinates. This allows to represent transformations as $3 \times 3$ matrices which can be incremented as the individual trees are processed. Every time a new cell is added to the tiling, this transformation translates and rotates a copy of the fundamental cell accordingly. Additionally, in each iteration step the parent polygon adjusts the transformation which created itself and passes it to the next polygon.

In order to generate tilings without duplicates, it is important that vertices are properly *closed*, meaning that no further copies are generated by rotations around a vertex once all $q$ adjacent cells at this vertex already have been constructed. The first *child* polygon of a *parent* polygon closes its leftmost vertex of that *parent* polygon. Hence, this polygon will create one *child* less and we call it a *closing polygon*. This is accomplished using a variable called *exposure*. It denotes the number of polygons left to be created by a specific *parent* polygon. In the regular case, it is given by $N_{\exp} = p - 2$, where a factor of $-1$ stems from the *parent* polygon and another factor of $-1$ from the adjacent closing polygon, which will be created by a *sibling*. For closing polygons, the exposure is reduced to $N_{\exp} = p - 3$, due to the connection to the formerly mentioned polygon.

In HYPERTILING we implement two variants of the improved version of Dunham's combinatorial algorithm, as outlined in Ref. [93] and a series of unpublished notes around the year 2007. The `DUN07` kernel represents an almost literal Python implementation, which suffers from a number of obvious performance bottlenecks. We address these in an improved version, named `DUN07X`, where we are able to achieve a significant performance speed-up and further-

more make the kernel ready for numba just-in-time compilation. Due to the recursive structure and hence independent trees, this kernel also offers a great starting point for parallelization, which we leave to future work.

A particular strength of Dunham's algorithm is that the number of polygons per layer as well as the correct parent-child relations are deduced exactly. However, it does not provide a trivial way of determining the neighbor relations during construction since the individual trees grow independently. A coordinate representation of cell vertices is still required to store and work with the tiling, introducing floating numbers and limiting the accuracy in practice.

The shortcomings with respect to the determination of neighborhood relations, the performance loss due to the construction of an entire tiling, as well as the recursive structure, which in general provides inferior performance compared to iterative sequential approaches, inspire us to develop the *generative reflection* (GR) kernel. It shares several strengths of Dunham's approach, such as an exact combinatorial scheme, which avoids duplicates and automatically determines when a layer is completed, but at the same time provides several substantial improvements. This includes a non-recursive algorithmic design, which exploits the discrete rotational symmetry of a regular tiling and allows to determine adjacency relations natively.

## 5.5 Generative Reflection Kernel

The *generative reflection* kernel (GR), in contrast to the other kernels available, entirely focuses on edges instead of vertices. To be specific, polygons are reflected on their edges rather than rotated around their vertices. The reflection process can be expressed as a sequence of Möbius transformations (compare Section 4.1).

Moreover, compared to the SRS kernel, which only creates one symmetry sector of the tiling and replicates this sector, the GR algorithm goes one step further and stores only the fundamental sector in the first place. All remaining cells (outside of this sector) can readily be constructed on-demand using Python generators. This way both the construction time, as well as the memory footprint can be reduced dramatically, even compared to Dunham's algorithm, as will be compared in Section 5.6.

The GR kernel provides a number of methods that hide the generative nature from the user, satisfying the usual interfaces, as previously touched upon in Section 3.1. For instance, if a polygon is accessed, the kernel will determine whether it is stored inside the fundamental sector (and therefore physically stored in the memory) or whether it needs to be generated first. Either way, an array containing center and vertex coordinates is returned.

### 5.5.1 General Concepts

While kernels such as SRS and SRG check for duplicate polygons globally, the GR kernel is a local combinatorial algorithm similar to DUN07X. In order to create a new cell, only the immediate neighborhood structure, but no knowledge of the remaining lattice is required. The creation of duplicates is avoided from the beginning. In the following, we sketch the algorithmic approach of how this is achieved:

A polygon is represented as an array $v$ of vertices $v_i$, stored in clock- or counterclockwise order, hence defining a new property: the *orientation*. In Figure 12, the orientation and start vertex (i.e. the first vertex in the array) are indicated by arrows. Given a specific *parent* polygon (black square), for the *children* (red, green, and yellow), which are created through edge reflections, the orientation changes, and the position of the starting points, relative to the parent, shifts. However, we require the relative orientation and starting point to match that of the parent polygon and hence adjust both properties. In Figure 12, on the left-hand side, the orientation is not corrected. Considering an arbitrary child (green, yellow, and red), the orientation (circular arrow) is different in relation to the parent. Moreover, considering

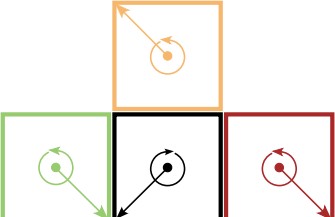 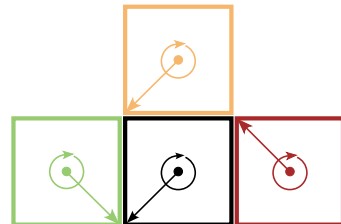

Figure 12: A parent polygon (black) and its children, constructed by edge reflections. Left: Orientations and starting points of the children are different from the parent. Right: Orientations and starting points of the children have been adjusted.

all the children, the starting point, i. e. the first vertex in the array (indicated by the straight arrow) is different with respect to the parent for each but the first child. For the first child (green), the starting point and the following vertex are shared with the parent. For the next child (yellow) however, the starting point, as well as the following vertex, are not shared with the parent. The right-hand side of the figure shows the adjusted polygons. Each shares the starting point and their last vertex with their parent. Moreover, their orientation is corrected to be clockwise. In summary, the orientation in each child is flipped and shifted by $i$, where $i$ describes which edge, counting from the starting point it has been reflected on.

Hence the orientation is now similar for every polygon and those edges which can create further children can be deduced. As shown in Figure 12, it is always the last edge (relative to the orientation pointer) which is the one that is shared with the parent. Moreover, polygons do not share an edge with their siblings, unless $q = 3$, where this is the case for the first and the second-to-last edge. Only those edges which are not shared with either the parent or a sibling can be used to generate another *child* in the next layer. However, for certain polygons, which we will refer to as *filler polygons*, this picture is not sufficient. Filler polygons are the equivalent to closing polygons in Dunham's kernel (compare Section 5.4). However, for GR a distinction between two types of filler polygons needs to be made. In comparison to regular (i. e. non-filler) polygons, one additional edge is blocked by either a parent or a nibling. A further distinction is made between filler polygons of first and second kind, illustrated in the left and right panel of the Figure 13, respectively. Filler polygons of the first kind are created through a child-nibling artifact. This describes the situation when a child of a polygon at a certain edge has already been created as a nibling. In other words, a *child* of a *sibling* can also be a *child* of *self*. Filler polygons of the second kind are not created through this effect, but instead, a *child* and a *nibling* are neighbors. As filler polygons of the first kind can be created by two parents, in order to prevent duplicates, *self* has to verify that the last created *nibling* is not a neighbor of *self*. In this case, *self* will create one child less, i. e. will not create the child of the first edge. For filler polygons of the second kind, the *child* created first is compared to the last polygon created by the parent immediately before the current parent. If both share an edge, they are considered filler polygons of second kind, and the corresponding edges are blocked.

Depending on whether the lattice parameter $q$ is odd or even, regular tilings feature either both or only one kind of filler polygon. This can be understood as follows. Naturally, the first set of vertices to be closed are those of the fundamental polygon. Specifically, for each vertex, $q - 1$ additional polygons are required. As the polygons are created pairwise on the two open edges connected to a vertex, for $q$ being odd, a final *pair* of polygons will close this vertex. However, the vertex in between them, i. e. the one on the edge they share, starts with two polygons instead of one. Therefore, it needs to be closed like the even-$q$ case for the fundamental polygon. In this case, the final polygon can be created by both open edges. The polygon thus has two parents and is therefore a filler polygon of first kind. Its vertices behave

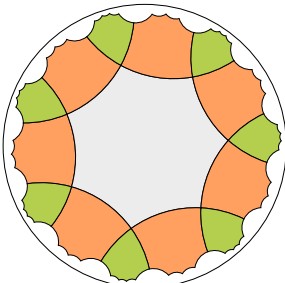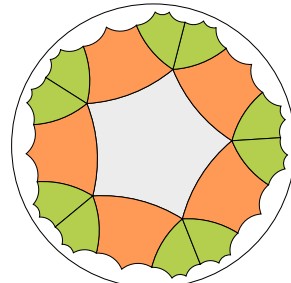

Figure 13: Left: Filler polygons of first kind (green) and non-filler polygons (orange) in the second layer of a $(7, 4)$ tiling. Right: Filler polygons of second kind (green) in a $(5, 5)$ tiling.

like the vertices of the fundamental polygon. Hence, for $q$ even, the tiling consists of regular and filler polygons of the first kind only. However, for $q$ odd, the tiling consists of regular polygons and alternating types of filler polygons. For $q = 3$, a different scenario arises. In this case, each polygon is connected to two of its siblings and thus every polygon is a filler of second kind. However, while for $q \neq 3$ filler polygons of second kind are only connected to one single other filler polygon of second kind, for q = 3 the filler polygons of second kind are connected to two other filler polygons of second kind. Together these polygons act as two separate pairs whereas each pair creates a filler of first kind in the next layer. These first kind filler polygons are also connected to their siblings and thus can be interpreted as filler polygons of both types, i.e. first and second kind, at the same time.

As cells are created in strictly deterministic order, i. e. layer by layer and in counter-clockwise direction, all filler polygons involve the last created *nibling* and the first *child* of the next polygon. If the first created *child* is equal to the last created *nibling* the filler polygon is of first kind. If the last created *nibling* is a neighbor to the first *child*, both are filler polygons of second kind. Therefore, the vertices of each first *child* of a polygon are compared to the edges of the immediately preceding one. If two consecutive vertices match, the polygons share an edge and for both polygons, the corresponding edge is blocked for a *child*. Whether an edge is blocked or not is an important quantity in this algorithm and is encoded as a single bit.

### 5.5.2 Algorithmic Details

In Algorithm 1, we present the core construction principle of the GR kernel as a pseudo code. Variables are highlighted using italic letters. In order to improve the readability of the code, certain passages have been simplified by means of helper functions. In the actual implementation, these sections are directly incorporated into the main function in order to avoid extra function calls and enhance runtime performance. The designated helper functions are:

- BLOCK_PARENTS_EDGES: For the array *edge_array*, which encodes whether edges of polygons are blocked or free, the entry for every polygon is an unsigned integer considered as a bitset, where each bit represents a certain edge of the respective polygon. For the initial representation, every bit except the bit for the parent's edge is set to 1. The corresponding integer is given as $\eta = 2^p - 1$, where $p$ is the number of polygon edges, as usual.

- CALCULATE_TARGET_LAYER_SIZE: Resort to analytical function in order to determine target layer size. In the actual implementation, this is done in advance of the function call in order to allocate a sufficient amount of memory.

- CHECK_FILLER_1ST_KIND: Checks whether the first input argument is a filler polygon of first

---

**Algorithm 1** Pseudo code of the GR kernel construction algorithm.

1: **function** GENERATE($p$, $q$, $n_{\text{layers}}$, *polygons*)
2:
3:    ▷ *Instantiate an array to record which edges are blocked*
4:    *edge_array* ← BLOCK_PARENTS_EDGES($p$)
5:
6:    ▷ *Iterate over layers to be constructed*
7:    **for** $i_{\text{layer}}$ **from** 0 **to** $n_{\text{layers}}$ **do**
8:        *layer_size* ← CALCULATE_TARGET_LAYER_SIZE($p$, $q$, $i_{\text{layer}}$)
9:
10:        CHECK_FILLER_1ST_KIND(*poly*, *polygons*, *edge_array*)
11:
12:        ▷ *Iterate over polygons in layers*
13:        **for** $j$ **from** *index_shift* **to** *layer_size* + *index_shift* **do**
14:            *poly* ← *polygons*[$j$]
15:            ▷ *Traverse edges and create child by reflection if not blocked*
16:            **for all** edges $e$ in *poly* **do**
17:                **if** $e$ is not blocked **then**
18:                    *child* ← REFLECT(*poly*, $e$)
19:                    CORRECT_ORIENTATION(*child*)
20:
21:                    **if** RADIUS_OF(*poly*) < RADIUS_OF(*child*) **then**
22:                        *polygons*[*poly_counter*] ← *child*
23:                        *poly_counter*++
24:                        CHECK_FILLER_2ND_KIND(*poly*, *polygons*, *edge_array*)
25:
26:                        **if** layer is completed **then**
27:                            **continue** with next layer
28:
29:        *index_shift* ← *index_shift* + *layer_size*

---

kind by comparing vertices of the next parent and the most recently created polygon. This is done using a $p \times p$ matrix $\eta$, where the components are defined as $\eta_{ij} = (v_i - \hat{v}_j)$, with $v$ and $\hat{v}$ representing the respective polygon vertices. The row index indicates the vertex of $v_i$ and the column index is the vertex of $v_j$ for a certain component. Now, $\eta_{ij}$ is compared against zero, with a threshold in order to account for numerical uncertainties. This way, matching vertices are identified. If two *consecutive* vertices match, a filler polygon of first kind is found and the indices in the matrix $\eta$ can be used to block the corresponding edge.

- CHECK_FILLER_2ND_KIND: Similar to CHECK_FILLER_1ST_KIND. Instead of the next parent, its first child is used.

- REFLECT: Computes vertex coordinates for a new polygon through reflection across a specific edge of the parent polygon. In practice, this is done by a series of Möbius transformations. To be more precise, the polygon is shifted in a manner that positions the initial vertex of the edge to be reflected upon at the center of the tiling, i.e. at $0 + 0i$. Subsequently, a rotation ensures that the second vertex of the edge under consideration is also located on the real axis. The actual reflection is done by a complex conjugation, which inverts the imaginary values of the vertices. Finally, the polygon is rotated again and moved back to its original position.

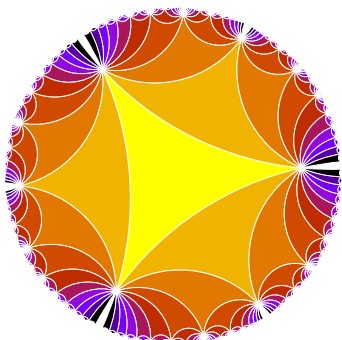 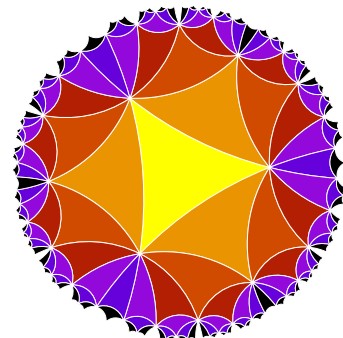

Figure 14: Left: (3, 20) grid with ten layers. A vertex shared by two regular polygons needs $\Delta i = q/2 = 10$ subsequent layers to be closed. Right: (3, 10) lattice with seven layers. Here, $\Delta i = 5$, hence only vertices of the first layer are closed.

- `CORRECT_ORIENTATION`: Ensures the correct sequence of vertices for the polygon, as illustrated in Figure 12.

- `RADIUS_OF`: Calculates the distance of the polygon's center from the origin as $r = \sqrt{z\bar{z}}$.

In the pseudo-code, input arguments are the lattice parameters $p$ and $q$, as well as the number of reflective layers to be constructed, $n_{\text{layers}}$. Lastly, the *polygons* variable represents an array that will serve as the storage for the polygons. It already contains the fundamental polygon. In the algorithm's progression, the actual offspring polygon is created in line 24 and inserted into the list of cells in line 27. The variables *poly_counter* and *index_shift* denote a global polygon number counter and a counter which accounts for the appropriate adjustment of indices when a new layer is initialized. They are initially set to 1 and 0, respectively.

### 5.5.3  Layer Definition

With the novel generation scheme of the GR kernel comes an adjusted definition of a *layer*, which is different compared to the traditional definition used in the other kernels of this package – the so-called *reflective layer*. It represents the natural layer definition for this kernel and is defined as the accumulated total number of reflections that need to be executed to construct a specific polygon. To be specific, let a parent be located in layer $n$, then its child, with whom the parent shares an edge, will be part of layer $n + 1$. This is different compared to the traditional layer definition where children are constructed by rotations about vertices and therefore might only share a vertex with the parent. We show examples for two different combinations of $(p, q)$ in Figure 16. In panel (a), cells are colored according to the traditional layer definition and in panel (b) according to their reflective layer attribute.

In general, it is important to remark that the parameter $n$ in the input parameter set $(p, q, n)$ denotes the reflective rather than the traditional layer number. For given $n$, a lattice generated by GR will therefore in general feature fewer cells than a lattice generated by other kernels. Due to the different definitions of layers (compare Section 5.5.3), also the overall shape of the tiling is generally different compared to other kernels. The layer definitions coincide for $q = 3$ as the vertices will be closed in every reflective layer, however for any $q > 3$, the behavior is different. In the traditional layer definition, boundary vertices are immediately closed once a new layer is constructed, i.e. all polygons which share this vertex are constructed and it becomes a bulk layer. In the GR, a boundary vertex in layer $i$ will be closed only

$$\Delta i = \begin{cases} \frac{q}{2} & \text{for } q \text{ even} \\ \frac{q-1}{2} & \text{for } q \text{ odd} \end{cases} \tag{31}$$

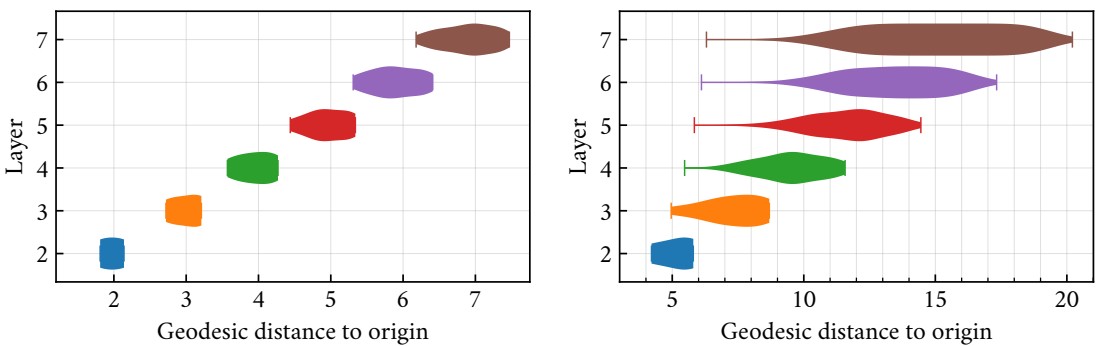

Figure 15: Distributions of cell distances to the origin for a $(7, 3)$ tiling (left panel) and a $(7, 20)$ tiling (right panel).

*reflective* layers later. Therefore, the vertex remains at the lattice boundary until $\Delta i$ further layers have been constructed. This holds for every vertex. Hence, as $q$ is increased, the tiling becomes increasingly sparse and its surface more and more fractal. Two examples are depicted in Figure 14. The vertices appear as white dots due to the boundary lines of the polygons. In the left panel, we show a (3,20) lattice with ten layers. The vertices for the first layer are not yet closed. In contrast, the vertices of the first layer in the right panel are closed. The vertices of the second layer, however, are still open, as one further layer is required to close them. Consequently, by design, since in the GR a lattice is built until the target number of reflective layers is reached, the completeness of the last $(q-3)/2$ traditional layers can not be guaranteed.

For compatibility reasons we provide functions that return the layer of a polygon in the traditional definition, `map_layers` and `get_layer`. The former method performs the actual computation, whereas the latter one represents a wrapper for convenient access. Note that upon the first call to `get_layer`, the mapping function is invoked internally, unless manually executed beforehand. Once mapped, subsequent calls are then simple memory access operations and hence fast.

It is clear that among the cells in a layer, the radial distances to the origin vary and, moreover, that the corresponding distance distribution of two adjacent layers in general overlaps. This is illustrated in Figure 15 for a (7,3) tiling (left panel), as well as for a (7,20) tiling (right panel). For $q = 3$, where the traditional and reflective layer definitions coincide, all open vertices of a layer are immediately closed by the subsequent one, which makes the overlap minimal. Instead, the overlap is considerably large for $q = 20$ since subsequent $\Delta i = q/2 = 10$ layers are required to close a vertex. The filler polygons in a layer always represent the maximal radial extension into previous layers.

### 5.5.4   Graph Kernels

One notable strength of the GR algorithm is its ability to deduce neighbor relations during the lattice creation process with only minor algorithmic adjustments. To this end, we provide specialized variants of the GR kernel, the *generative reflection graph* (GRG) and the *generative reflection graph static* (GRGS) kernel. For regular polygons and $q \neq 3$, the adjacency relations are entirely given by parent-child relations, i.e. we need to keep track of which child polygon is created by which parent. If the polygon is a filler of first kind, the polygon is connected to the next *sibling* of the *parent*. If the polygon is a filler of second kind, the polygon is connected to its next *nibling*. Due to the order of construction, these are either stored next to the *parent* (first kind) or *self* (second kind). For $q = 3$, the relations change such that the polygons in the

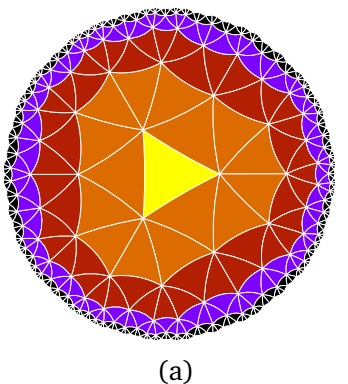 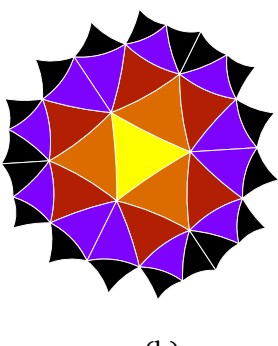

(a) (b)

Figure 16: Comparison of traditional (a) and reflective layers (b) for a (3,7) tiling. In both cases, $n = 5$, however the resulting lattice size (number of cells) is different. Only for $q = 3$, both definitions become identical (not shown).

array next to *self* are also neighbors to *self*.

As mentioned above, both kernels are based on the GR construction principles. However, due to their nature as *graph kernels*, only the neighbor relations and the centers of polygons are returned instead of entire polygons, which allows the kernel to be particularly lightweight. However, note that during the construction process, the coordinates of the vertices are still required, resulting in a larger memory footprint during this step compared to the final graph. As another consequence, lattices constructed by GRG and GRGS are still limited by the available computational accuracy, as coordinate calculations involve floating point arithmetics.

Unlike GRG and GR, the GRGS kernel stores adjacency relations of the entire lattice explicitly. This can be useful in applications, where a radial instead of sectorial ordering of the cell indices is more intuitive or preferred for other reasons, such as in simulation comprising radial boundary conditions.

### 5.5.5 Neighbors

As mentioned above, the GRG and GRGS kernels construct adjacency relations automatically. For the original GR kernel, this is not the case, however, we provide a number of methods that allow to obtain the neighbors of a polygon in a separate second step. The methods can be accessed either directly or using the `get_nbrs` method. The latter wrapper function, which is shared by all kernels (compare Section 3.3), takes a keyword `method` which specifies the algorithm. In the following, we go through all methods available in the GR kernel in detail:

- `get_nbrs_generative` or `method="GEN"`
  Neighbors are determined in a brute-force manner, where the center of the cell is reflected across all edges, and resulting points are compared to the center coordinates of every other cell in the lattice. This is achieved by calling the helper routine `find` on these points, which, as detailed earlier, extracts the corresponding polygon index. As a consequence, similar to the `find` function itself, execution times for this method heavily depend on the structure and size of the lattice. However, at the same time, this method is therefore relatively robust and good for testing and debugging purposes.

- `get_nbrs_radius` or `method="RAD"`
  This method calculates the overall distance between the considered polygon and all other polygons in a brute-force manner. In case the distance matches the lattice spacing, the respective polygons are considered neighbors.

- `get_nbrs_geometrical` or `method="GEO"`
  In `"GEO"` we compute adjacency relation using the *relative* positions of polygons inside their reflection layers, defined as

$$\Gamma_i = \frac{i}{N_j}, \tag{32}$$

  where $i$ is the index of the polygon and $N_j$ is the number of polygons in its corresponding reflection layer $j$. This measure can be interpreted as an *angular* position in the layer. Let us consider an arbitrary child-parent pair, where the parent index is $i$ and the child index $K$. By design, we know that the child is created directly from its parent and therefore the relative positions in their respective layers must be similar. This allows to provide a precise *guess* for the index of neighbor cells, given by

$$k \approx \Gamma_i \cdot n_k. \tag{33}$$

  It needs to be remarked that filler polygons create a minor additional shift. In practice, a small region around the guess index is scanned for children and as soon as one is found, the remaining ones can be determined straightforwardly. Only for $q = 3$, the *siblings* of a polygon are its neighbors. They are always located at indices $i - 1$ and $i + 1$.

- `get_nbrs_mapping` or `method="MAP"`
  This method assumes that the neighbors of all cells in the fundamental sector have been mapped (i.e. explicitly stored) earlier and are hence immediately available. This can be achieved by invoking the function `map_nbrs` once. In case the region of interest is outside the fundamental sector, all relations are properly adjusted. In case the neighbors have not been mapped beforehand, `get_nbrs_mapping` will perform the mapping first, resulting in a significantly larger execution time for the initial call. The way `map_nbrs` works is that it computes the distance of the current polygon to all polygons in the previous layer in order to find the two closest points which are candidates for its parents. Analogously, the next layer is searched for the remaining neighbors. In the process, all candidates are compared against the lattice spacing in order to determine which of them are actual adjacent cells. Only in case of $q = 3$, *siblings* are relevant, which can be readily identified as the immediately adjacent indices of *self*.

Additionally to these methods, which return neighbors of single cells, the GR kernel provides an implementation of `get_nbrs_list`, which returns the neighbors for the entire tiling. This function calls `map_nbrs` and generates a list format from the result. Be aware that compared to invoking `map_nbrs`, the memory requirement is larger by a factor of about $p$, since `get_nbrs_list` is not restricted to the fundamental sector but explicitly returns all connections in the lattice.

### 5.5.6   Integrity Check

One additional noteworthy feature of the GR kernel is the `check_integrity` routine, which, to some extent, allows monitoring the correctness and completeness of a grid. The test is carried out in two stages. At first, the tiling is checked for duplicates as well as for problems resulting from the available numerical resolution. To this end, a polygon is removed and the `find` method is employed. This method determines the index of the polygon a coordinate point is located in and returns `False` in case none is found. If *another* polygon is found at that location, this either indicates the existence of a duplicate or a spatial displacement of adjacent polygons due to numerical accuracy. Once this first test is passed successfully, we proceed by testing for the correct number of neighbors using the `get_nbrs_generative` method, which has been explained earlier. This method is specifically suited for this purpose as it does not search

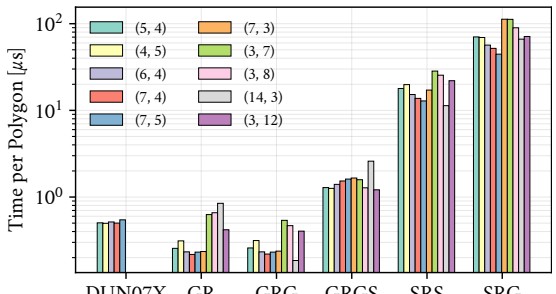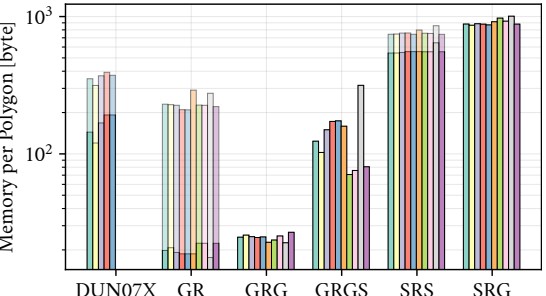

Figure 17: Computing time and memory footprint comparison for different kernels and $(p, q)$ lattices. In the right panel, peak memory consumption is displayed for the construction process (solid bars) and neighbor search (transparent bars). Kernels that compute neighbors during lattice construction present only solid bars.

for neighbors directly but creates the neighbors through reflections. Subsequently, the `find` method is used on these newly created polygons. If the number of found neighbors does not match $p$, the tiling either has holes or the polygon under consideration is located in a boundary layer. In either case, the method will generate a warning message containing the index of the first polygon lacking an adequate number of neighbors, along with its corresponding layer information. A code example can be found in the package documentation.

## 5.6 Benchmarks

In this section, we provide detailed performance results for tiling construction and the generation of adjacency relations. These benchmarks should aid the selection of the appropriate kernel when computing resources are a constraint. All tests have been performed on a desktop workstation with an Intel Xeon W-1390p and 64 Gb RAM. Besides these performance benchmarks, a feature-based comparison is presented in Section 5.7.

For a given $(p, q)$ lattice, we anticipate the kernels to generate polygons at a steady rate, with some additional algorithm-specific overhead which should be negligible for large tilings. This time per polygon is therefore a good indicator for the kernel performance. Results for different kernels are shown in the left panel of Figure 17. In the Figure, the heights of the individual bars represent the average time to construct a single polygon $t_{\text{poly}}(p, q, n)$ for a specific tiling. This quantity is determined using linear regression of the time required to construct a full tiling, $t(p, q, n)$ and the number of polygons $N(p, q, n)$. Therefore, in the regression the function

$$f : N(p, q, n) \in \mathbb{N} \rightarrow t(p, q, n) \in \mathbb{R}. \tag{34}$$

is fitted. A similar approach is used for calculating the memory footprint per polygon, displayed in the right panel of Figure 17. The advantage of a linear regression over a simple arithmetic average is the compensation of systematic shifts due to systematic overhead. Error bars are mostly smaller than 1% and not shown.

In general, GR and GRG are the fastest algorithms in the package. Remarkably, DUN07X' runtime is of the same order of magnitude as those of GR and GRG, even though it creates an entire tiling instead of only one sector. It is more than two times faster than GRGS, which is a variant of GRG that constructs a full tiling. The main reason for its performance is that DUN07X dispenses with explicit coordinate comparisons. The SR kernels are, as expected, significantly slower, even when only one sector is created, such as for SRS. This stems from the internal duplicate management system of the SR kernels, which even with its optimized containers and efficient binary search, introduces a systematic performance overhead for large tilings. Finally,

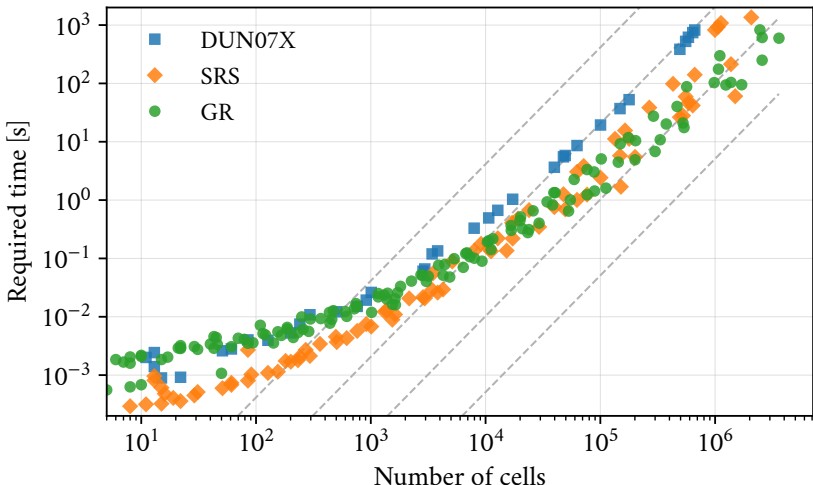

Figure 18: Computing time required for detecting adjacent cells. Graph kernels generate neighboring relations automatically and are hence not included. Dashed lines indicate quadratic scaling.

our analysis reveals that SRG offers the poorest performance, which is the expected trade-off for its dynamics manipulation capabilities.

In the analysis above we focus solely on the time required to generate the tiling. However, for many applications adjacency relations are essential and measurements encompassing the time both for tiling creation and for retrieving neighbor information become relevant. Considering graph kernels, GRG and GRGS are the fastest implementations in the package. As expected, GRGS is slower by roughly a factor of $p$ compared to GRG. We also find that once neighbor relations are required, DUN07X turn out to be one of the slowest kernels for large lattices, as the lack of specialized neighbor detection methods leads us to resort to a brute-force radius search instead. As the majority of adjacency algorithms do not scale linearly with the number of polygons, there is not a simple time-per-polygon ratio as in Figure 17 and we present our benchmark results separately in Figure 18. Besides DUN07X, we evaluate SRS and GR with their default algorithms `ROS` (see Section 5.3.4) and `get_nbrs_list` (which calls `MAP` and expands the results from one sector to the whole tiling), respectively. Despite numerical optimizations, they all exhibit quadratic scaling for large tilings, owing to the all-to-all distance comparisons needed. The GR kernel displays a larger overhead, but benefits from its heavily optimized internal structure and is generally the fastest for large lattices.

Another important performance quantity is the required memory per polygon. It is best evaluated by monitoring the *peak* memory consumption during the construction of tiling and neighbor structure, shown in the right panel of Figure 17 as solid and transparent bars, respectively. Again, linear regression is used to compensate for systematic offset. The results clearly show that GR and GRG heavily outperform every other kernel in this category, due to the fact that only one sector is stored and generators are used to construct remaining regions only on demand. As soon as neighbor relations are required, however, their memory footprint increases significantly, as an explicit, full list of neighbor indices is produced. The GR kernel nonetheless remains the most memory efficient option. Finally, SRS as well as SRG are the most expensive kernels in this category, stemming from the same memory requirement for the list output combined with algorithmic overhead due to the `HyperPolygon` class and the duplicate management mentioned earlier.

## 5.7 Choosing a Kernel

For many use cases, any of the kernels available gets the job done, and the user does well to stick to the default options. For those that require specific features, however, selecting the most suitable kernel among the currently available options, each with its own distinct technical intricacies, may seem challenging. In this section, we give some guidance to help making this choice.

The first distinction to be made is between graph and tiling kernels. Graph kernels are more lightweight and contain mostly only the neighbor relations, whereas geometric information is reduced to a minimum. In contrast, tiling kernels store the entire tiling, including all vertex coordinates, which obviously requires additional memory capacity. The graph kernel category currently contains the GRG and GRGS kernels, both in the GR family. The key difference between these two is that GRG only creates one sector explicitly and thus requires less memory. On the other hand, GRGS generates a full grid, which leads to an increased generation time but offers certain advantages, such as a more natural, radial ordering of cell indices instead of sectorial ordering. This facilitates a simple integration of GRGS into simulations that require radial boundary conditions, for instance.

The tiling kernel category also contains a member of the GR family, the GR kernel, which stands out in terms of performance and significantly improved memory efficiency compared to the family of SR kernels. The latter comprises the kernels SRS and SRG, both of which offer greater flexibility than the GR kernel, enabling dynamic modifications of a lattice in SRG and the use of refinements in SRS. Another difference between them is the calculation of neighbors, which is already done during the creation of the grid in the SRG kernel. Hence they are promptly available, without any additional function calls, as for the graph kernels mentioned above. With respect to overall performance, SRS benefits from a fast, sector-based construction approach, whereas the dynamic manipulation properties of SRG require more advanced internal bookkeeping, leading to some performance overhead.

For a concise overview, see the summary of the various kernel properties in Table 3. Note that our plotting and animation functions work with all kernels, since they share a common API. We also want to remark that the summary in the Table represents a current depiction, and as the library continues to evolve, some kernels may receive further updates and additional features in the future.

| | SRS | SRG | DUN07X | GR | GRG | GRGS |
|---|---|---|---|---|---|---|
| Construction principle | sector | full | full | sector | sector | full |
| Neighbors built during construction | ✗ | ✓ | ✗ | ✗ | ✓ | ✓ |
| Vertex coordinates available | ✓ | ✓ | ✓ | ✓ | ✗ | ✗ |
| Dynamic modifications | ✓ | ✓ | ✗ | ✗ | ✗ | ✗ |
| Refinements | ✓ | ✗ | ✗ | ✗ | ✗ | ✗ |
| Integrity checks | ✗ | ✗ | ✗ | ✓ | ✓ | ✓ |
| Numerical performance (no neighbors) | ● | ● | ●● | ●●● | ●●● | ●● |
| Memory efficiency (no neighbors) | ● | ● | ●● | ●●● | ●●● | ●● |
| Memory efficiency (with neighbors) | ● | ● | ●● | ●● | ●●● | ●●● |

Table 3: Feature comparison of the construction kernels in the package. More bullets correspond to better performance.

# 6 Examples

In this section we offer a selection of scientific uses of the HYPERTILING package.

## 6.1 Epidemic Spreading

The spreading of infectious diseases can be modeled by simple dynamic rules, such as the so-called contact process [96], in more technical terms also referred to as the *asynchronous susceptible-infected-susceptible* (SIS) model. Regardless of any immediate application of epidemic spreading in hyperbolic geometry [97], it can be seen as an instance of a more general class of models, the so-called directed percolation (DP) universality class [96, 98, 99]. In nature, DP is realized in a number of phenomena, such as forest fires [100], catalytic reactions [101], interface pinning [102] and turbulence [103]. Hence, the SIS model simulated in this section serves the purpose of investigating a broader range of systems.

Sticking to the epidemic language, on a lattice structure, each vertex represents an individual that can be in one of two states, *infected* (active) or *healthy* (susceptible). The temporal evolution of the system comprises two fundamental stochastic processes [104]. Denoting an active site by $A$ and a susceptible site by $\varnothing$, these are infection of a neighbor, $A \rightarrow A + A$, and spontaneous recovery, $A \rightarrow \varnothing$. Both processes run stochastically with rates determined by the infection probability $\lambda$. Over time, a *cluster* of infected sites evolves. If the infection spreads slowly (small $\lambda$), the system is dominated by the recovery process and eventually an *absorbing* state is reached where the entire lattice is inactive and the disease is extinct. In contrast, if $\lambda$ is large enough, the system steadily maintains an active cluster and is said to be in the *active phase* where the disease persists indefinitely. Typically, at a critical parameter $\lambda_c$ whose precise value depends on the microscopic details of the lattice, there is a transition between the active and the absorbing phase. Specifically, at $\lambda = \lambda_c$ spatial and temporal correlation length scales, $\xi_\perp$ and $\xi_\parallel$, diverge and the emerging activity cluster becomes scale-invariant and strikingly self-similar [50, 105]. If the lattice structure is regular or – in a certain sense – only weakly disordered [106], the properties of this transition are *universal* and fall into the class of *directed percolation* [107].

We conduct epidemic spreading simulations on a hyperbolic (5,4) tiling with 120 million cells, starting from a single infected seed particle in an otherwise inactive lattice. The seed is positioned at the center of the tiling. Throughout the time evolution of the contact process, we monitor the number of active sites $N_a(t)$. Our goal is to locate the critical point, i. e. the probability threshold $\lambda_c$ which separates the active from the inactive regime. Results are displayed in Figure 19. As configurations of individual runs typically vary greatly, we average each curve over up to 25 000 independent realizations of the process, i. e. we measure the ensemble averages $\langle N_a(t) \rangle$ as well as the survival probability $\langle P_s(t) \rangle$, given by the fraction of runs which are still alive at time $t$. The results indicate that the threshold probability is located between $\lambda = 0.625$ and $\lambda = 0.626$. The cluster size $\langle N_a(t) \rangle$ for smaller values $\lambda$ declines to zero, after an initial transient behavior, whereas for larger values of $\lambda$, the curves display exponential growth. A similar picture emerges in the second panel of Figure 19, where the survival probability $\langle P_s(t) \rangle$ decays to zero below the threshold and tends to constant values above, indicating the infection persists on the lattice, i. e. an active phase.

In all simulations we monitor the mean square geodesic radius of the spreading cluster and stop the simulation once this quantity reaches the order of magnitude of the lattice boundary region in order to avoid finite-size effects, since at this point, the process can not spread further. Moreover, we use random sequential updates, i. e. in every time step an active particle is randomly selected and updated according to the dynamic rules described above. Then, time is incremented by the inverse of the current cluster size, $1/N_a$.

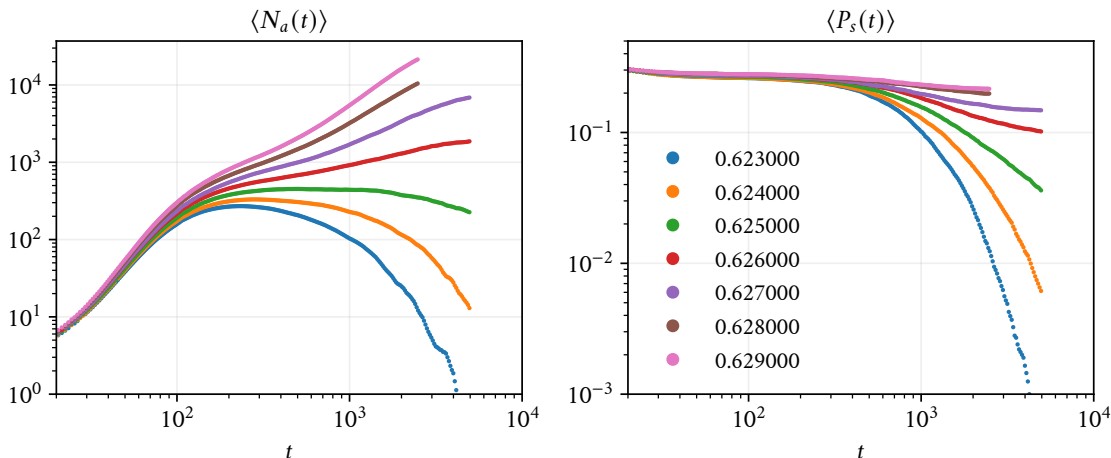

Figure 19: Epidemic spreading on a large hyperbolic (5,4) tiling. The left panel shows the time evolution of the number of active sites, whereas the right panel displays the corresponding survival probabilty. Colors indicate the infection probability.

Concluding this section, we want to emphasize that state-of-the-art high precision simulations of reaction-diffusion processes, such as in the context of epidemic spreading from a localized seed, require considerably large lattices in order to avoid boundary effects and to extract the true asymptotic scaling behavior. Since the dynamics of continuous phase transitions in non-Euclidean geometry is generally a challenging field of research (see Section 1.2 and references therein), which requires substantial resources when addressed numerically, a more detailed study falls out of the scope of this paper and is left to future work.

## 6.2   Scalar Field Theory

In quantum field theory, one of the most fundamental nonlinear models is the $\phi^4$-model [108], a scalar field with fourth-order self-interaction, described by the Lagrangian

$$\mathcal{L} = \frac{1}{2}\partial_\nu \phi \partial^\nu \phi - \frac{1}{2}\mu^2 \phi^2 - \lambda \phi^4 \tag{35}$$

in Minkowski spacetime, where the parameter $\mu$ is associated to the mass and the partition function (or generating functional) is given by

$$\mathcal{Z} = \int \mathcal{D}\phi \, e^{iS} = \int \mathcal{D}\phi \, \exp\left(i \int \mathrm{d}^n x \mathcal{L}\right). \tag{36}$$

The phase structure of this model resembles that of an Ising model [109, 110], a well-known model in statistical mechanics and magnetic solids. In particular, a critical transition from a high-temperature disordered phase into an ordered low-temperature phase where the symmetry of the scalar field is spontaneously broken, can be observed. Simulations of this theory on a lattice first require a suitable Wick rotation, transforming the path integral factor $e^{iS}$ into a Boltzmann exponent $e^{-S_E}$, thus resulting in a classical $n$-dimensional Euclidean theory. The actual lattice discretization is straightforward and described in many textbooks, such as [111, 112]. Eventually, one finds a lattice Hamiltonian of the form

$$\beta \mathcal{H} = S_E = -J \sum_{\langle i,j \rangle} \phi_i \phi_j + m \sum_i \phi_i^2 + \lambda \sum_i \left(\phi_i^2 - 1\right)^2, \tag{37}$$

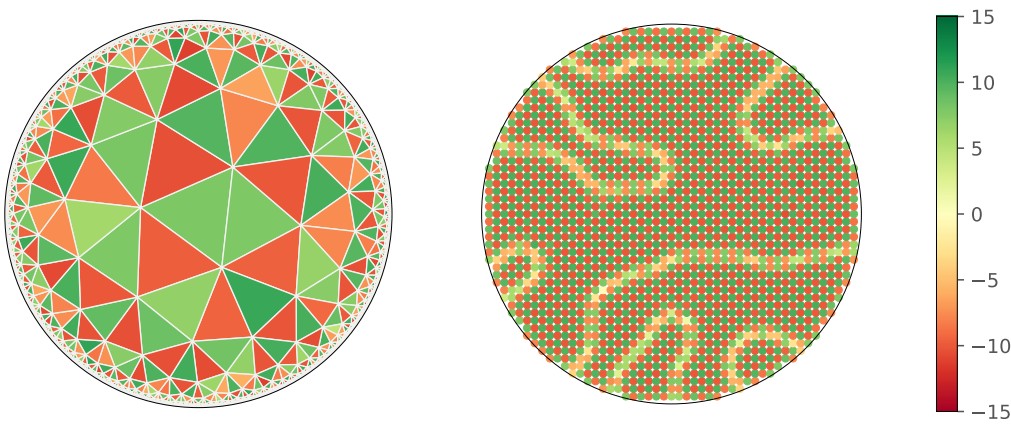

Figure 20: A quenched antiferromagnetic spin model exhibits geometrical frustration on a (3,7) tiling (left panel), whereas on a flat lattice, an ordered anti-parallel alignment can be observed (right panel). Colors indicate the field values.

where $\phi_i$ is an $N$-component real variable and the angular brackets $\langle i, j \rangle$ denote a summation over nearest neighbors with coupling $J$. The inverse temperature is denoted by $\beta$. For any positive $\lambda$, this system undergoes a continuous phase transition which lies in the universality class of the classical $O(N)$ model, which can be seen as the $N$-component vector-valued generalization of the Ising model. The parameter $\lambda$ proves particularly useful in numerical analysis since it can often be adjusted in a way that leading scaling corrections approximately vanish. In this case, one speaks of an *improved* Hamiltonian [113, 114]. For $\lambda \to \infty$ the classical Ising, XY, Heisenberg, and higher–symmetry vector–models are recovered, as the field is effectively forced to unit-length $\phi_i^2 = 1$.

In order to illustrate how a remarkably distinct behavior can arise on hyperbolic geometries, compared to flat Euclidean lattices, we simulate the classical scalar-field Hamiltonian on two different lattice structures, a (3,7) lattice, as well as a flat square grid, both confined to a circular region. We choose the couplings between adjacent cells ("spins") in a way that energetically favors anti-parallel local alignment ($J < 0$). Both systems are initiated in a random hot configuration of positive (red) and negative (green) field values, which – in the magnetic picture – can be interpreted as uniaxial spins pointing upwards or downwards, respectively. By repeatedly applying the Metropolis update algorithm [115] the system eventually reaches thermal equilibrium. An animation of this process is available on our YouTube channel[5]. In Figure 20 we show examples of equilibrium configurations for both systems. In the traditional flat lattice, we find extended anti-ferromagnetic domains, separated by so-called anti-phase boundaries [116, 117], originating from the quench of the initially hot system towards a colder temperature. In the hyperbolic lattice, however, imposed by connectivity restrictions, local forces always compete with each other, and a true ground state (a perfect anti-parallel alignment) cannot exist. Instead one finds what is commonly referred to as geometrical frustration [118].

## 6.3 Helmholtz Equation

The hyperbolic graph structure generated by HYPERTILING can be used to construct discrete lattice operators and solve partial differential equations. For example, the discretized Helmholtz

---

[5]https://www.youtube.com/watch?v=Sh49whM4CZA

operator for a general graph can formally be written as a matrix $w(A - G) + mI$ acting on a vector of length $N$, in our case corresponding to scalar function values of $N$ cells in the tiling. Here, $A$, $G$ and $I$ denote adjacency, degree and identity matrices, respectively, each of size $N \times N$. While $w$ denotes appropriate geometric weights as discussed in Ref. [78] for hyperbolic tilings, $m$ represents the "mass" term, also referred to as eigenvalue $\lambda$ or wave number $k^2$ of the Helmholtz equation.

We provide interfaces to compute these matrix operators once the neighbor relations of a `HyperbolicTiling` or `HyperbolicGraph` have been obtained (compare Sections 3.1 and 3.2) as demonstrated in the following code fragment:

```python
import hypertiling as ht

# create tiling and obtain neighbours in one step
nbrs = ht.HyperbolicTiling(7,3,6).get_nbrs_list()

A = ht.operators.adjacency(nbrs)   # Adjacency matrix
G = ht.operators.degree(nbrs)      # Degree matrix
I = ht.operators.identity(nbrs)    # Diagonal mass matrix

D = A - G # Laplacian
H = D - M # Helmholtzian
```

As hyperbolic tilings quickly can become very large, a memory-efficient sparse matrix format is used instead of a dense representation. In Figure 21(a) we display the Helmholtz operator for a $(7,3)$ tiling with 3 layers, where we set $w = 2$ and $m = 5$. As can be seen, the adjacency matrix is indeed sparse and furthermore symmetric due to the graph being undirected. The sevenfold pattern can be directly related to the construction kernel, in this case, SRS, where only one symmetry of the tiling is explicitly created and then replicated. Note that both constant and position-dependent weights can be set for all operators using the optional `weights` keyword (not shown in the code fragment above). Moreover, the operator functions accept an optional `boundary` keyword which takes a boolean array of length $N$ (number of cells), encoding whether a vertex is considered a boundary vertex. For boundary points, the corresponding rows are left empty, the matrix is non-symmetric. This characteristic provides an approach to Dirichlet boundary value problems, since this way it is ensured that boundary points act as *ghost cells*; they affect their bulk neighbors, but remain unchanged themselves. The right-hand side of the resulting linear system thus carries precisely the Dirichlet boundary values for those sites.

Now we are ready to solve the linear system $Hx = y$, where $H$ is the differential operator, $x$ is the solution vector and $y$ is the right-hand side. We use GMRES [119, 120], which is an established solver for non-symmetric and potentially indefinite linear systems. In Figure 21(b), we show an example where we solve an electrostatic boundary value problem in a $(3,7)$ tiling with 6 layers and one refinement step (compare Section 3.4). The boundary has been fixed to values of either $-1$ (red) or $+1$ (blue). In this example, we set $m = 0$, hence reducing the Helmholtz equation to a Laplace problem. In the interior region of the lattice a smooth "electrostatic potential" is obtained, as expected, with interfaces between areas of positive and negative solutions roughly following hyperbolic geodesics. A more detailed exploration of Helmholtz and Laplace problems on hyperbolic tilings can be found in our example notebooks, provided in the HYPERTILING repository.

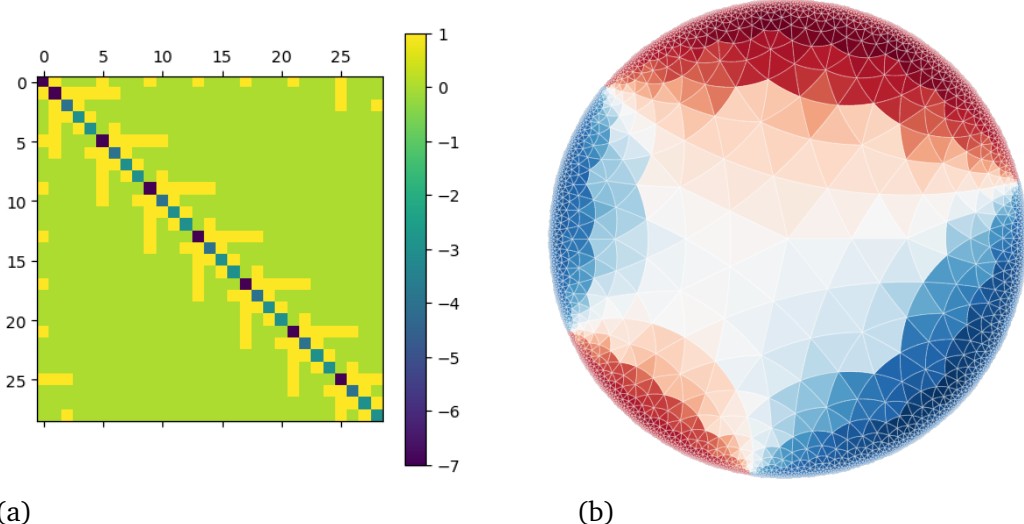

(a)                                                              (b)

Figure 21: Left: Discretized Helmholtz operator for a (7,3) tiling with 3 layers. Right: Solution of an electrostatic problem on a refined (3,7) tiling, where boundary values have been fixed to either -1 (red) or +1 (blue).

## 7  Road Map

With its wide range of scientific applications and potential for producing visually captivating artistic content, hyperbolic geometry remains an active and evolving field. The discretization of hyperbolic Riemann surfaces, in particular, grows in importance across a number of scientific disciplines. This continued interest motivates us to further extend HYPERTILING's capabilities and we already have some new features in the pipeline, some of which we outline below.

### 7.1  Triangle Groups

The HYPERTILING package focuses on *regular* tilings, given by the $(p,q)$ Schläfli symbol. As explained above, these represent maximally symmetric spatial discretizations, which makes them particularly amenable to (large-scale) numerical simulations. However, tessellations of Riemann surfaces with polygons realized by so-called triangle reflections group [32, 121] can be expanded beyond regularity. A triangle group is defined by three rational numbers $(p,q,r)$ which reciprocally encode the inner angles at their vertices, given by $\pi/q$, $\pi/q$ $\pi/r$. Depending on the relation $1/p + 1/q + 1/r$ of the fundamental *Schwarz triangle* [122] being larger, equal, or smaller than one, the surface is respectively spherical, flat, or hyperbolic. From the fundamental triangle, we can build a tesselation through reflections on its edges. It is clear that there are $2p$, $2q$, and $2r$ triangles meeting at the corresponding vertices. The smallest triangle group lattice is given by $(7,3,2)$. Note that $p$, $q$ and $r$ can become formally infinite, representing tilings with triangles that have one or more ideal vertices (compare, e. g., Figure 6 in Section 4). Regular $(p,q)$ tilings can be recovered by coalescing $2p$ triangles in a $(p,q,2)$ triangle group tiling.

### 7.2  Regular Maps

Given the generically large boundary of finite hyperbolic tilings, compactified periodic lattices can prove themselves useful in many applications, particularly in computational physics, where one often wants to keep the influence of simulation boundaries under control. Translationally invariant compact hyperbolic tessellations can be constructed using regular maps [45] and we

intend to tackle the challenge of implementing this possibility in future releases. Currently, the user may, for instance, use the dynamic features of the SRG kernel (compare Section 5.3) to manually construct periodic tilings by identifying edges of a properly chosen fundamental domain, as outlined for example in References [43] and [48].

### 7.3 Symbolic Kernels

As mentioned in Section 5.5, the GRG kernel comprises a combinatorial algorithm that is able to automatically compute neighborhood relations. This local construction principle allows to entirely move away from coordinates representations of cells and to construct the graph relations solely from its combinatorial principles. We are currently implementing this as a variant of the GR kernel family, where we also intend to encode all polygon states and transformations as elementary bit operations. The corresponding smaller memory demands and overall increase in efficiency should allow for the creation of even larger tilings. In fact, as tilings constructed through this symbolic approach are necessarily exact since no floating point arithmetic is involved, there is no demand for exponentially increasing numerical precision. Hence, lattices of arbitrary size can be generated, only restricted by available computing resources. This approach shows similarities with more group-theoretic symbolic algorithms, where tilings are processed using a representation of triangle group elements as words produced by finite-state automata [123–126].

### 7.4 Parallelization

So far we do not offer *parallel* lattice construction in HYPERTILING. One reason for this is that most of our kernels are designed for efficient serial execution. In the GR kernel, for the generation of every new polygon, the precise state of the previously generated one is required. Therefore exactly only one polygon can be created in each iteration. The static rotational kernels SRS and SRG on the other hand use global duplicate management containers (compare Section 5.3.3) which need to be accessed frequently and hence would require heavy inter-node communication in a distributed memory environment. Alternatively, a single node shared-memory parallelization, where the lattice is simultaneously expanded by several replication drivers, each working through a share of the currently outmost polygons to be replicated, would be possible. Also, the DUN07 algorithm offers potential for parallelization, given its tree-like structure, where independent sub-trees can be traversed in parallel. However, as the calculations inside a single tree do not contain many operations of the same type, it is questionable whether the use of a GPU is efficient here or whether bypassing the global interpreter lock by means of running several processes on the same CPU might be superior. A detailed consideration is reserved for future work.

Finally, we would like to note that, although there are potential performance gains from parallel implementations of some of the existing algorithms, this would only yield notable advantages for extremely large sizes, due to the already considerable single-core speed of the existing serial implementations. Moreover, in situations involving numerical simulations on large tilings, almost always the actual simulations are bound to consume substantially more resources than the lattice creation process, when done with the current version of HYPERTILING. Hence, parallelization is not one of our top priorities at the moment.

## 8 Conclusion

In this article, we present an open-source library for the construction, modification and visualization of regular hyperbolic tilings. We developed HYPERTILING to give researchers and visual

artists access to hyperbolic lattices by using straightforward high-level Python commands. Our primary emphasis lies on high-performance computing, and our algorithms are optimized to deliver the highest possible speed and memory efficiency. To further support scientific applications, our library includes an extensive tool set of methods to establish adjacency relations, enabling the use of tilings as graphs. In this context, our generative reflection (GR) kernel family currently is, to the best of our knowledge, the fastest available algorithm to construct these geometric structures, substantially outperforming other standard algorithms. This performance can be achieved by employing a combinatorial design of the algorithm, bit-coded memory layout, generator functionality (to significantly decrease storage requirements) as well as Python optimizations such as numpy and numba's just-in-time compilation. Given these optimizations, the GR kernels are able to construct huge lattices – a crucial requirement for scientific applications where large hyperbolic bulk regions with minimal boundary effects are needed. For instance, a $(7, 3)$ lattice with 1.5 billion cells (roughly equivalent to a $35\,000^2$ square lattice) can be generated on a standard desktop workstation in less than one hour.

In addition to speed, the library also enables flexible and dynamic lattice manipulation by means of its static rotational (SR) kernels: A tiling can be created explicitly layer-wise or expanded around particular cells, and cells can be added or removed at any place in the lattice. This is made possible by a sophisticated bookkeeping system that avoids the creation of duplicate cells. Together with fast, optimized implementations of all required Möbius transformations provided in the package, this opens the way for the construction of very individual, even procedural and dynamically generated lattices. The SR kernels also provide triangle refinements, which can be particularly useful whenever a better spatial resolution beyond the geometrically fixed hyperbolic polygon edge length is required. Finally, a Python implementation of the well-known construction algorithm by D. Dunham is available, both in its original, as well as in a modern, performance optimized version.

All internal arithmetics and algorithmic details are hidden from the user. The interested developer can nevertheless profit from a hierarchy of class objects, which allows easy debugging, modification and, most importantly, extendability.

Our library provides considerable plotting and animation capabilities. We offer plotting methods that are both fast as well as accurate, and tilings can moreover be exported as SVG vector graphics for further manipulation and publication-ready plots. Finally, our animation classes facilitate the presentation of dynamic processes on a hyperbolic lattice as well as changes of the lattice itself, all realized via straightforward matplotlib animation wrappers.

The combination of all these resources renders HYPERTILING a uniquely capable package. It puts large hyperbolic lattices at the fingertips of researchers, lowering the threshold for exploration and opening up new avenues of application of numerical hyperbolic geometry.

## Acknowledgements

We thank P. Basteiro, J. Erdmenger, H. Hinrichsen, R. Meyer, A. Stegmeier and L. Upreti for fruitful discussions. We are also grateful to the Information Technology Center of Würzburg University for providing computational resources through the JULIA cluster.

**Author contributions** M.S. initiated and leads the project. M.S. and Y.T. conceived the core program design and implemented the main lattice construction algorithms. F.G. specialized on numerical optimization and HPC aspects of the code. F.D, F.G, D.H. and J.S.E.P. implemented specific modules and extensions and tested the code. All authors contributed to writing the documentation and the manuscript.

**Funding information**    M.S. acknowledges support by the Deutsche Forschungsgemeinschaft (DFG, German Research Foundation) under Germany's Excellence Strategy through the Würzburg-Dresden Cluster of Excellence on Complexity and Topology in Quantum Matter - ct.qmat (EXC 2147, project-id 390858490). F.G. furthermore acknowledges financial support through the German Research Foundation, project-id 258499086 - SFB 1170 'ToCoTronics'.

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
