# Peer review of "HYPERTILING -- a high performance Python library for the generation and visualization of hyperbolic lattices"

_SciPost Physics Codebases, doi:SciPost Phys. Codebases 34-r1.3 (2024) , SciPost Phys. Codebases 34 (2024)_

## Round 1 · Referee Report · Anonymous (Referee 1) · 2024-2-5

Strengths

1- Highly optimized Python library for hyperbolic tessellations running on a workstation. 2- Easy to download and install the Python libraries. 3- The possibility to generate a variety of regular lattices within the Poincare disk. 4- High-performance plotting and animations with dynamic lattice manipulations. 5- Entirely mathematical and physical backgrounds provided at a sufficient level.

Weaknesses

1- A small typo in Eq. (1), where the last term in the hypersurface matric $x^2_2$ should be $x^2_n$.

Report

I am pleased to find such a useful tool (software) to generate the regular hyperbolic lattices on the Poincare disk which has been needed. It is amazing to what extent the user can generate and modify the hyperbolic lattices, mark the polygons by colors, and perform various measurements. To my present knowledge, I have not met any software of such complexity and usefulness for the scientific community.

Questions about whether the parallelization on CPUs and GPUs was satisfactorily discussed. Physical motivations were given, too. From a mathematical point of view, I found the results correct. The Python-generated plots were well-presented, accurately analyzed and concisely discussed. For all these reasons I approve the current manuscript for publication.

Requested changes

Apart from the minor typo in Eq. (1), no further corrections are necessary.

---

## Round 1 · Referee Report · Anonymous (Referee 2) · 2024-2-11

Strengths

-Library is written in Python-- hence easily accessible to users

-High-performance optimizations and a variety of algorithms have been implemented

Weaknesses

-The paper provides just an overview of the capabilities of the library.
For an actual use in a project an in-deep documentation would be needed.

Report

The manuscript presents a Python library for hyperbolic lattices, a discretization of two-dimensional space with constant negative curvature.
Hyperbolic tilings have attracted a lot of interest in many domains of physics. Yet, their generation requires considerable technical knowledge and so far a generic, publicly available library was missing.
The present contribution is therefore very welcome.

The manuscript gives a short quick setup, discusses the main features of the library, and summarizes the mathematical framework relevant for hyperbolic tilings.
This is followed by a more technical overview of the kernels and algorithms.
A few examples and some perspectives on future developments end the manuscript.

In my opinion the manuscript generally satisfies the criteria for acceptance.

My main criticism is that the manuscript gives just an overview of the features of the library, citing the "package documentation" for more details.
For a concrete use of the library, a user would probably need these more details.

I presume that this style of presentation has been chosen to avoid an overly long and technical manuscript.
Yet, I would encourage the authors to provide a more structured documentation, for example by providing a hierarchy of the classes (if there is such a hierarchy) and tables of the available methods of the classes, options for the factory methods etc.
This would give a more focused description of what the library can actually do, without referring to an external documentation.

Related to this, a link to the package documentation should be provided in the text.

Some minor points:

-Eqs. (1) and (2): there is probably a typo in the index (n is not defined, also in Eq. (1) the last index is probably not 2)

-Quick start: I think it should be useful to already mention what is the meaning of the parameter n.

-What would be the motivation of studying epidemic spreading on a hyperbolic lattice?

-Eq. (37): I think in general there should be a coupling constant in front of the nearest-neighbor interaction.
Shouldn't the discretization of the action (35) always lead to a ferromagnetic lattice model?
The example below refers to an antiferromagnetic model, where the coupling constant would be of different sign than in Eq. (37).

---

## Round 2 · Referee Report · Anonymous (Referee 2) · 2024-6-20

Report

The authors have positively answered my remarks, and in particular have added a technical section describind the general structure of the library.
Thus I recommend the publication.

I still do not understand eq. (1) and (2). Is it perhaps n=d+1 there?
If so, this should be corrected.

Recommendation

Ask for minor revision

---

## Round 2 · Author Response

Dear Referees,

Thank you for your careful reading and helpful comments.

Regarding the main criticism on one of the reports, over the need for more detailed documentation:

We opt to have a full code documentation as part of the repository, in order to - as indeed already noted by the referee - avoid an overly long and technical manuscript. This documentation is continuously improved, concomitantly with the code, as the package evolves. Also, providing extensive code details in the manuscript would risk it quickly becoming outdated, as the package developing continues.

Nevertheless, in the revised version of the manuscript we provide a new section (5.1) which addresses the code structure from a high level perspective, employing a uml-type diagram which shows the hierarchy of available kernel classes. We also now explicitly emphasize the availability of an external documentation in the Quick Start section.

Regarding the minor points raised in the report:

  • thank you for pointing out the typo in Equation 1
  • we added an explanation for the parameter "n" in Section 2.3
  • we added some clarification on the relevance of studying epidemic spreading on hyperbolic spaces at the beginning of section 6.1
  • we added the coupling constant to the Hamiltonian in Equation 37 and mentioned that it is set negative in the example we give

Best regards, Manuel Schrauth

---

## Round 2 · List of Changes

The revised version of the manuscript contains a new section (5.1) which addresses the code structure from a high level perspective, employing a uml-type diagram which shows the hierarchy of available kernel classes. We also now explicitly emphasize the availability of an external documentation in the Quick Start section.

Minor changes:

  • we corrected a typo in Equation 1
  • we added an explanation for the parameter "n" in Section 2.3
  • we added some clarification on the relevance of studying epidemic spreading on hyperbolic spaces at the beginning of section 6.1
  • we added the coupling constant to the Hamiltonian in Equation 37 and mentioned that it is set negative in the example we give

---

## Round 3 · List of Changes

Typos in Equations (1) and (2) have been fixed.

---

## Editorial Decision

published